JCB Journal of Cell Biology

# Structural characterization and inhibition of the interaction between ch-TOG and TACC3

James Shelford[1]*, Selena G. Burgess[2]*, Elena Rostkova[3], Mark W. Richards[2], Gabrielle Larocque[1], Josephina Sampson[2], Christian Tiede[2], Alistair J. Fielding[4], Tina Daviter[5], Darren C. Tomlinson[2], Antonio N. Calabrese[2], Mark Pfuhl[3], Richard Bayliss[2], and Stephen J. Royle[1]

**The mitotic spindle is a bipolar array of microtubules, radiating from the poles which each contain a centrosome, embedded in pericentriolar material. Two proteins, ch-TOG and TACC3, have multiple functions at the mitotic spindle due to operating either alone, together, or in complex with other proteins. To distinguish these activities, we need new molecular tools to dissect their function. Here, we present the structure of the α-helical bundle domain of ch-TOG that mediates its interaction with TACC3 and a structural model describing the interaction, supported by biophysical and biochemical data. We have isolated Affimer tools to precisely target the ch-TOG-binding site on TACC3 in live cells, which displace ch-TOG without affecting the spindle localization of other protein complex components. Inhibition of the TACC3–ch-TOG interaction led unexpectedly to fragmentation of the pericentriolar material in metaphase cells and delayed mitotic progression, uncovering a novel role of TACC3–ch-TOG in maintaining pericentriolar material integrity during mitosis to ensure timely cell division.**

## Introduction

Chromosome segregation during mitosis is driven by the mitotic spindle (McIntosh, 2016). To form a spindle, the centrosomes separate and move to opposite ends of the cell, and a bipolar microtubule array is generated between them. The microtubules contact the kinetochores of each paired sister chromatid to coordinate their movement, while at the other end, the microtubules are focused at the spindle pole which contains a centrosome: a centriole pair embedded in pericentriolar material (Prosser and Pelletier, 2017). Understanding how the mitotic spindle assembles and operates is a major goal in cell biology.

Many proteins contribute to the formation and function of the mitotic spindle, and this paper focuses on two: ch-TOG/ CKAP5 and TACC3. These proteins interact with each other and participate in a number of activities (Saatci and Sahin, 2023). First, a non-motor protein complex that binds spindle microtubules (MTs) is composed of TACC3, ch-TOG, clathrin, and GTSE1 (Fu et al., 2010; Hubner et al., 2010; Lin et al., 2010; Booth et al., 2011; Bendre et al., 2016). This complex stabilizes the bundles of spindle microtubules that attach to kinetochores by physically crosslinking them (Hepler et al., 1970; Booth et al., 2011; Nixon et al., 2015, 2017). The complex is formed by clathrin and TACC3 at its core, which bind to the microtubule lattice, while ch-TOG and GTSE1 bind, respectively, to TACC3 and

clathrin as ancillary subunits (Hood et al., 2013; Burgess et al., 2018; Ryan et al., 2021). Second, ch-TOG can bind to kinetochores, independently of microtubules, via an interaction with Hec1, where it plays a role in mitotic error correction (Herman et al., 2020). Third, TACC3 and ch-TOG, in an exclusive complex, track the growing ends of microtubules in the spindle (van der Vaart et al., 2012; Nwagbara et al., 2014; Gutiérrez-Caballero et al., 2015). Fourth, at the spindle pole, ch-TOG localizes to the centrosome whereas TACC3 is observed close by Gutiérrez-Caballero et al. (2015). Understanding the relative contributions of these proteins to spindle assembly and function is therefore complicated and is hampered by a lack of molecular tools to deconvolute the roles of individual proteins cleanly.

Mammalian ch-TOG is a member of the XMAP215 family of microtubule polymerases. These proteins vary in size, with an N-terminal region comprising two, three, or five TOG domains and a C-terminal region that is variable. A TOG domain is a module consisting of six HEAT repeats that can bind to tubulin dimers (Ayaz et al., 2012; Fox et al., 2014). The current model for XMAP215 proteins is that they track the growing end of the microtubule and contribute to its polymerization using the multiple TOG domains (Brouhard et al., 2008). The variable C-terminal region in ch-TOG likely contains a cryptic sixth TOG

[1]Centre for Mechanochemical Cell Biology, Warwick Medical School, University of Warwick, Coventry, UK;   [2]School of Molecular and Cellular Biology, Astbury Centre for Structural Biology, Faculty of Biological Sciences, University of Leeds, Leeds, UK;   [3]School of Cardiovascular and Metabolic Medicine and Sciences and Randall Centre, King's College London, Guy's Campus, London, UK;   [4]Centre for Natural Products Discovery, School of Pharmacy and Biomolecular Sciences, Liverpool John Moores University, Liverpool, UK;   [5]Institute of Cancer Research, Chester Beatty Laboratories, London, UK.

*J. Shelford and S.G. Burgess contributed equally to this paper.   Correspondence to Stephen J. Royle: s.j.royle@warwick.ac.uk;   Richard Bayliss: r.w.bayliss@leeds.ac.uk; Mark Pfuhl: mark.pfuhl@kcl.ac.uk.

domain and a further small helical domain (Hood et al., 2013; Burgess et al., 2015; Rostkova et al., 2018). TACC3 is a member of the transforming acidic coiled-coil (TACC) family of proteins that each have a long coiled-coil region in their C-terminus, which is expected to govern their homodimerization (Peset and Vernos, 2008). TACC3 is a substrate of Aurora-A kinase with phosphorylation of a residue in the ACID region permitting the interaction with clathrin (Burgess et al., 2018). The TACC3–ch-TOG interaction is evolutionarily conserved. Examples include Alp7–Alp14 (yeast), TAC-1–Zyg9 (nematode), d-TACC–Msps (fly), and maskin–XMAP215 (frog). Mutational analysis has mapped the interaction between TACC3 and ch-TOG to a stutter in the TACC3 coiled-coil (residues 678–688) and a folded region (residues 1932–1957; C-terminal to TOG6) in ch-TOG (Hood et al., 2013; Burgess et al., 2015; Gutiérrez-Caballero et al., 2015; Rostkova et al., 2018). However, the structural details of this interaction are not yet determined.

The interplay between ch-TOG and TACC3 at spindle poles is particularly unclear. In *Drosophila*, D-TACC concentrates Msps at the centrosome (Lee et al., 2001). In vertebrates, the interaction between TACC3 and ch-TOG is thought to recruit the γ-tubulin ring complex (γ-TuRC) to the centrosome (Singh et al., 2014; Rajeev et al., 2023). XMAP215 can nucleate MT growth from MT seeds and γ-TuRCs, with the C-terminal region binding to γ-tubulin and N-terminal TOG domains stimulating nucleation (Thawani et al., 2018). Related to this, ch-TOG was shown to recruit γ-TuRC and activate it in interphase (Ali et al., 2023). Whether or not TACC3 is involved in this activity is unclear. In human cells, the localization of ch-TOG at mitotic centrosomes is independent of TACC3, which seems to be located distal to the centrosome (Gutiérrez-Caballero et al., 2015). In agreement with this, evidence from mammalian oocytes—where the spindle is formed without centrosomes—is that TACC3 forms a liquid-like domain at spindle poles (Fu et al., 2013; So et al., 2019). Finally, back to *Drosophila*, recent work indicates that D-TACC forms a liquid-like scaffold in the pericentriolar material (PCM) and not at the centrosome itself (Wong et al., 2025).

In this paper, we describe the structural details of the TACC3–ch-TOG interaction and report the discovery of Affimers that target the ch-TOG binding site on TACC3. The binding of Affimers occludes this site such that ch-TOG cannot associate with TACC3 in live cells. We use these inhibitory tools to demonstrate that the TACC3–ch-TOG interaction is required for PCM integrity during mitosis in human cells.

## Results

To produce a detailed model for the TACC3–ch-TOG interaction, biophysical and structural analyses of both proteins, alone and in complex, were carried out. Human TACC3 and ch-TOG constructs were expressed in *E. coli* with an N-terminal His-NusA-tag to improve soluble protein expression and purified via affinity-chromatography. The expression tag was removed by TEV cleavage followed by ion-exchange and size-exclusion chromatography to ensure high-sample purity for subsequent analysis.

### TACC3 is a parallel coiled-coil dimer

To determine the oligomeric state of the TACC domain of TACC3, analytical ultracentrifugation (AUC) experiments were performed on TACC3 629–838. Sedimentation velocity traces were fitted well for all cells with one predominant species observed at 1.4 S. The frictional ratio was ~2.2 but varied depending on the prevalence of an additional small species which appeared to reduce the frictional ratio in proportion to its abundance (Fig. S1, A–C). The high frictional ratio is consistent with the interpretation that TACC3 629–838 is an extended molecule: a rod shape with potentially unfolded parts at either end. The molecular weight in solution, calculated from sedimentation co-efficient and frictional ratio, is ~48–50 kDa, indicating that the protein is dimeric. A very small amount of a heavier species was present in all samples, which might represent a tetramer, but as it displayed no concentration dependence, it may be a disulfide-crosslinked protein (Fig. S1, A–C).

Next, to ascertain the orientation of the dimeric TACC3 TACC domain, electron paramagnetic resonance (EPR) spectroscopy was performed. MTSL-labeling was carried on native cysteine residues in TACC3 629–838 C749A, C828A and TACC3 629–838 C662A, C749A mutants, generating TACC3 MTSL-C662 and TACC3 MTSL-C828, respectively. Continuous-wave EPR spectra of TACC3 MTSL-C662 and MTSL-C828 (Fig. S1 D) showed a very slight broadening in comparison to one another, in support of a short interspin label distance at the upper limit of applicability between 1.6 and 1.9 nm (Banham et al., 2008). The corresponding DEER traces were weakly resolved, and it was not possible to extract a reliable distance measurement (Fig. S1, E, and F), consistent with an interspin distance at the lower range of borderline region of applicability. The presence of a dipolar interaction, although not well-defined in these experiments, supports TACC3 being a parallel dimer, where we would expect the labels to be in close proximity. These conclusions are consistent with existing X-ray crystal structures of the TACC domain fragment (TACC3 758–838, PDB 5LXN and 5LXO) in which this shorter region of the TACC domain is a dimeric, parallel coiled-coil.

### Structural characterization of the ch-TOG C-terminal domain

We have previously shown that the C-terminal domain of ch-TOG (residues 1517–1957) is sufficient for robust binding to the TACC3 TACC domain (Hood et al., 2013). Nuclear magnetic resonance (NMR) spectroscopy studies with the C-terminal domain (residues 1591–1941) from the *Drosophila* homolog of ch-TOG, minispindles (Msps), identified two independently mobile folded domains connected by a ~10 amino acid highly flexible linker. The N-terminal subdomain (residues 1591–1850) was characterized as an additional sixth TOG, and the C-terminal subdomain (residues 1860–1941) as an α-helix bundle (Hood et al., 2013; Burgess et al., 2015). The equivalent regions are retained in ch-TOG: the sixth TOG domain maps to residues 1517–1802 and the C-terminal α-helix bundle to residues 1817–1957.

In vitro co-precipitation studies with truncated ch-TOG constructs showed that the ch-TOG C-terminal α-helix bundle (1817–1957) is sufficient for binding to TACC3 (Fig. 1 A). We

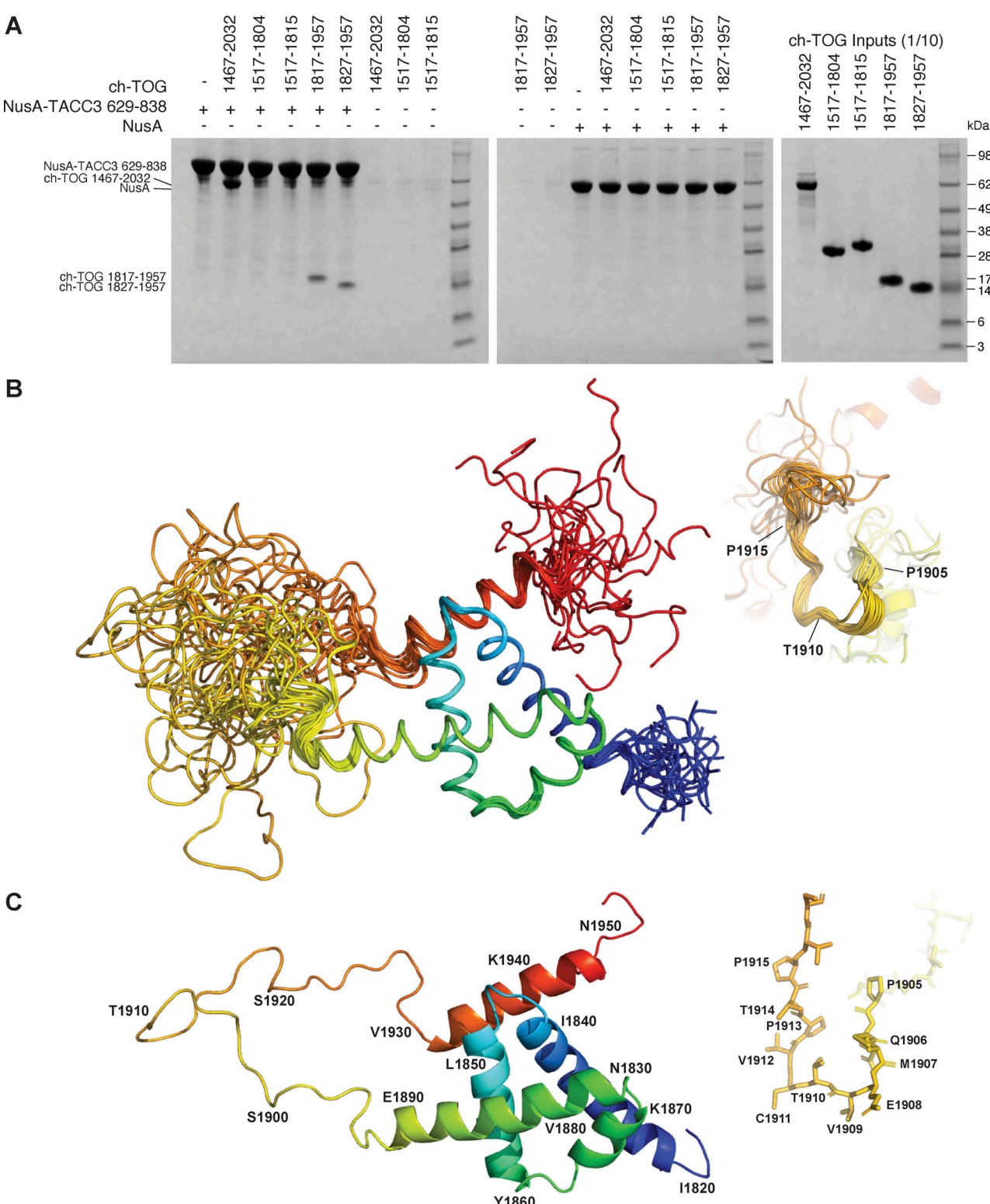

Figure 1. **Characterization of ch-TOG 1817–1957. (A)** In vitro coprecipitation assays between immobilized NusA-TACC3 TACC domain and ch-TOG truncates. **(B)** Superposition of the top 20 ch-TOG 1817–1957 NMR structures using the structured core of helices, H1-H4 (residues 1826–1894) for alignment. Inset, structures aligned on ch-TOG residues 1905–1915. **(C)** Cartoon representation of the best NMR structure for ch-TOG 1817–1957 is shown in the same orientation as in B. Inset, stick representation of ch-TOG residues 1905–1915. Structures in B and C are colored by spectrum mode where the N-terminus is blue and the C-terminus is red. Source data are available for this figure: SourceData F1.

**Table 1. Structure statistics & quality indices**

| | |
|---|---|
| **Total NOEs** | **3,322** |
| H-bond constraints | 52 |
| Dihedral constraints | 174 |
| RDCs | 93 |

proceeded to determine the structure of ch-TOG 1817–1957 by NMR spectroscopy. Spectra collected from stable isotope-labeled ch-TOG protein allowed full backbone and side-chain assignment (BMRB access code: 27235) (Rostkova et al., 2018). Structure determination was carried out using a combination of experimental NMR data comprising dihedral angle constraints extracted from backbone (HN, HA, CA, CB, C′) chemical shifts using DANGLE (Cheung et al., 2010), and distances from $^{15}N$ and $^{13}C$ resolved 3D NOESY-HSQC spectra and residual dipolar coupling constants (RDCs) (Table 1).

The best 20 NMR structures for ch-TOG 1817–1957 display good agreement within structurally ordered regions when superimposed (Fig. 1 B). The best individual structure for ch-TOG 1817–1957 is shown in Fig. 1 C. The domain consists of five helices, H1: 1825–1841; H2: 1846–1860; H3: 1865–1871; H4: 1874–1892, and H5: 1932–1945. Residues at the N- and C-terminus of the domain, 1817–1821 and 1946–1957, and the long linker region between H4 and H5, corresponding to residues 1898–1931, are disordered (Fig. S2 A).

The overall structure of the core domain (H1–H4) is composed of two pairs of V-shaped antiparallel α-helical hairpins connected by short loops. The V-shapes are created by α-helices having residues with small side chains proximal to the connecting loop and larger side chains nearer the other end. For the H1–H2 hairpin, these are G1849 at the narrow end, I1836 and I1853 in the middle, and V1829, L1833, and Y1856 at the open end (Fig. 2 A). The H3–H4 hairpin has S1872 and S1873 near the loop and I1865, L1869, and L1884 near the open end (Fig. 2 B). These V-shaped hairpins are rotated by ~90° relative to each other and stacked flat on top of each other, exchanging hydrophobic contacts that hold the two hairpins together, and form the core of the domain: H1 residues, V1829, L1833, I1836, F1837, and I1840; H2 residues, L1850, L1853, Y1854, and Y1856; H3 residues, I1865, F1868, and L1869; and H4 residues, F1876, V1880, L1884, and I1887. This hydrophobic core is devoid of polar amino acids and thus expected to be very stable (Fig. 2 C). By contrast, the interface with H5, which packs across the narrow end of the H1–H2 hairpin, is of a more mixed character, consistent with an unstable association; contacts are made between Y1935, L1939, and L1942 from H5, and K1839, E1835, K1838, S1842, N1845, and E1848 of the core domain (Fig. 2 D). The interaction may be further stabilized by the potential of salt bridges between residues R1938 and E1848, R1945 and E1844, R1943 and E1835, K1839 and E1835, E1855 and R1891, K1857 and E1881, and K1858 and E1888 (Fig. 2).

An intermediate structural state is observed for the short segment of residues P1905–P1915. This portion of the domain has very high RMSD values (Fig. S2 A). However, the distribution of φ/ψ values shows a more defined conformation (Fig. S2 A) with only

modest variation, and superposing the structures using this region shows a good agreement of the structures (Fig. 1 B inset). This is supported by higher heteronuclear NOE values compared with the rest of the H4–H5 linker (Fig. S2 B). This portion of the domain also responds strongly to H5 binding to TACC3 (Fig. 3 A, see below).

## Conformation and stability of ch-TOG 1817–1957

Residues in the core domain of the helix bundle (ch-TOG 1817–1957 H1–H4) have very low backbone RMSD values ~0.2 Å, large numbers of medium- and long-range NOEs, large positive heteronuclear NOEs, large ΔCA secondary chemical shifts, and sizeable RDCs, indicating a tightly packed, well-folded and highly rigid region of the protein (Fig. S2). Within this region, only the loop region between H2 and H3 displays higher RMSD values (~0.5 Å). Data corresponding to the N- and C-termini and the long H4–H5 linker (Fig. 1 B) are consistently in agreement with unstructured characteristics: large RMSDs, very few medium and long-range NOEs, negative or small heteronuclear NOEs, small CA secondary chemical shifts, and very small RDCs (Fig. S2). H5 exhibits intermediate RMSD values of 0.3–0.4 Å (Fig. S2) and appears to be in an equilibrium between a stable α-helical state, in which it is associated with the rest of the domain, and a dissociated state in which it is partially unfolded.

## Interaction of ch-TOG 1817–1957 with TACC3

TACC3 629–838 Δ699–765 was used for interaction studies with ch-TOG because this construct retains the binding properties of the complete TACC domain but displays improved biophysical characteristics making it more amenable to experimental analysis. TACC3 629–838 forms a gel-like precipitate at relatively low concentrations and reversibly precipitates at room temperature. These characteristics are mitigated by the deletion of residues 699–765. NMR titrations were performed with $^2H/^{15}N$ labeled ch-TOG 1817–1957 to which TACC3 629–838 Δ699–765 was added at a ratio of 1:3, respectively (Fig. S3 A). All peaks corresponding to H5 disappeared upon binding to TACC3, with substantial chemical shift perturbations at positions neighboring H5 (Fig. 3 A). Modest chemical shift perturbations were also seen in the region around I1840, which were involved in the association of H5 with the H1–H4 core, and the short proline-rich segment in the middle of the linker that connects H4 to H5. These results indicate that only H5 (residues 1932–1945) of ch-TOG is involved in binding to TACC3. This is in agreement with previous work where the deletion of ch-TOG 1932–1957 abolished binding to TACC3 (Gutiérrez-Caballero et al., 2015). As evidenced by the disappearance of H5 signals, while all other resonances remain visible, H5 is dislodged from the H1–H4 core concomitant with its binding to TACC3 (Fig. 3 A). Given the key role of ch-TOG H5 for complexation with TACC3, we tested whether a synthetic peptide corresponding to the H5 sequence retained the features required for interaction. The binding of the H5 peptide to TACC3 was measured with a $K_D$ of 17.3 ± 2.0 μM (Fig. 3 B).

## Model of the TACC3–ch-TOG complex

A structural model of the interface between TACC3 and ch-TOG was generated by supplying AlphaFold2-Multimer (Evans et al.,

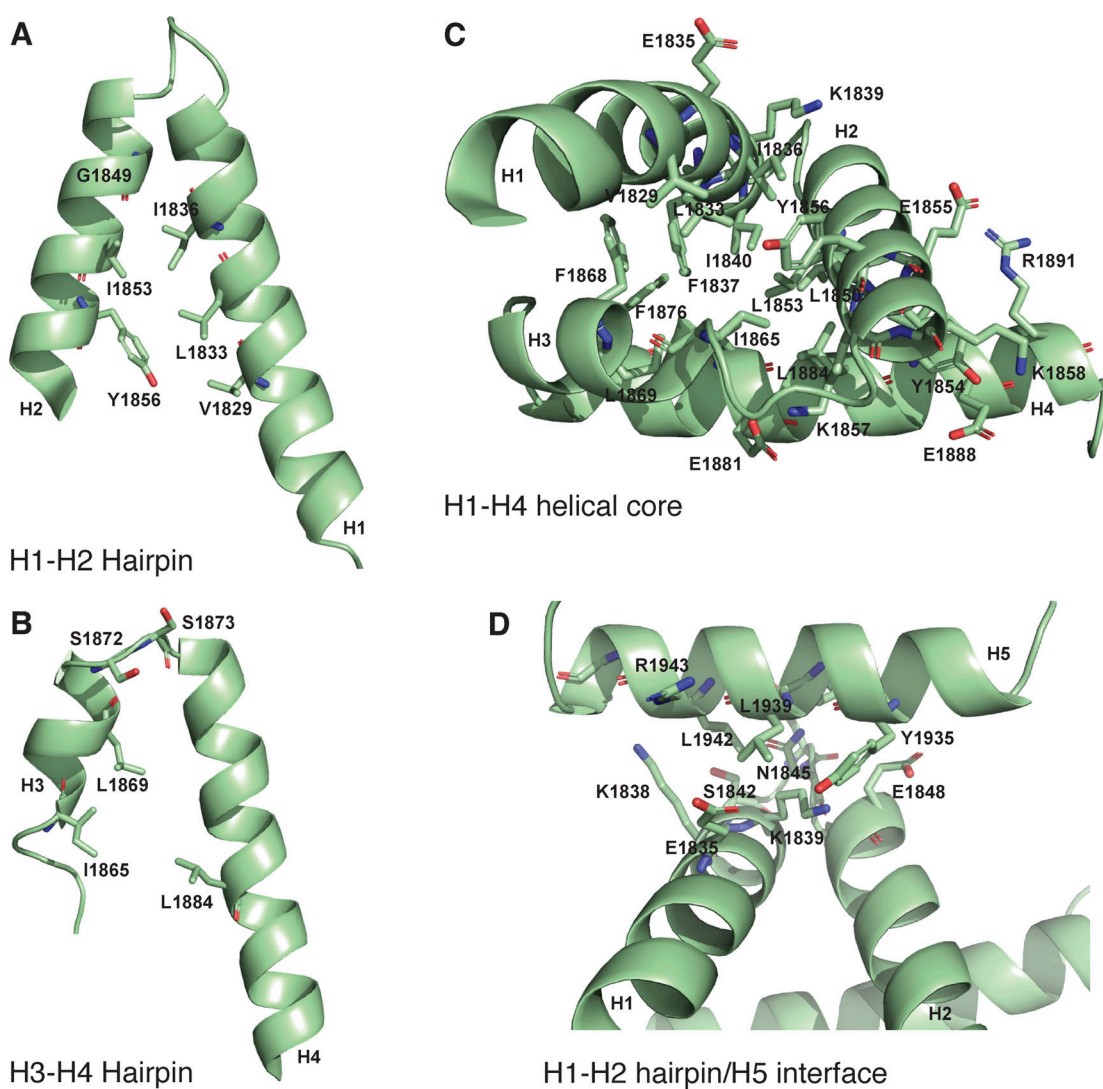

**Figure 2. Structural interactions within ch-TOG 1817–1957. (A)** Cartoon representation of the helical hairpin formed between H1 and H2. **(B)** Cartoon representation of the helical hairpin formed between H3 and H4. **(C)** Cartoon representation of the H1–H4 helical core. **(D)** Cartoon representation of H5 contacts with the H1–H2 hairpin.

2021, *Preprint*; Jumper et al., 2021; Mirdita et al., 2022) with two copies of the sequence of TACC3 629–838 Δ699–765 and one copy of ch-TOG 1817–1957. Consistent with our AUC and EPR spectroscopy data, and with existing crystal structures, this region of TACC3 is predicted with high confidence as a dimeric parallel coiled-coil. The structure predicted for ch-TOG 1817–1957 is consistent with our NMR data (Fig. 1 B): a core bundle of four α-helices is formed by residues 1817–1892, and a fifth α-helix formed by residues 1931–1946 (H5) is separated from the bundle by a long, disordered region. Rather than folding back onto the bundle, H5 is confidently predicted to be flipped out and interacts with the TACC3 dimer (Fig. 3 C and Fig. S3 B). Consistent with the NMR data (Fig. 3 A), the interface with TACC3 is formed exclusively by H5, the core α-helical bundle having no consistently or confidently predicted involvement. H5 binds into the groove between the two protomers of the TACC3 dimer in the region formed by residues 672–687. This binding site is

consistent with previous observations that deletions of residues 678–681 or 682–688 of TACC3 abolished the binding of ch-TOG (Hood et al., 2013). The interface creates a short stretch of trimeric coiled-coil with the three amphipathic α-helices packing hydrophobic side chains together to form its interior (Fig. 3 D). The principal hydrophobic sidechains contributed by ch-TOG are those of Leu1939 and Leu1942, which were previously identified as critical residues for the interaction (Gutiérrez-Caballero et al., 2015) (Fig. S3 C), while Leu672, Ile675, Phe679, and Val682 are contributed by TACC3 protomer A, and Met676 and Val683 by protomer B. Outside of the coiled-coil, intermolecular salt bridges appear to be formed between Arg1943 of ch-TOG and Glu681 of TACC3 protomer A, and between Arg1938 of ch-TOG and Asp677 of protomer B. Mutation of ch-TOG Arg1938 and Arg1943 results in a reduction in binding between ch-TOG and TACC3 supporting the role of these residues in complex formation (Fig. S3 C).

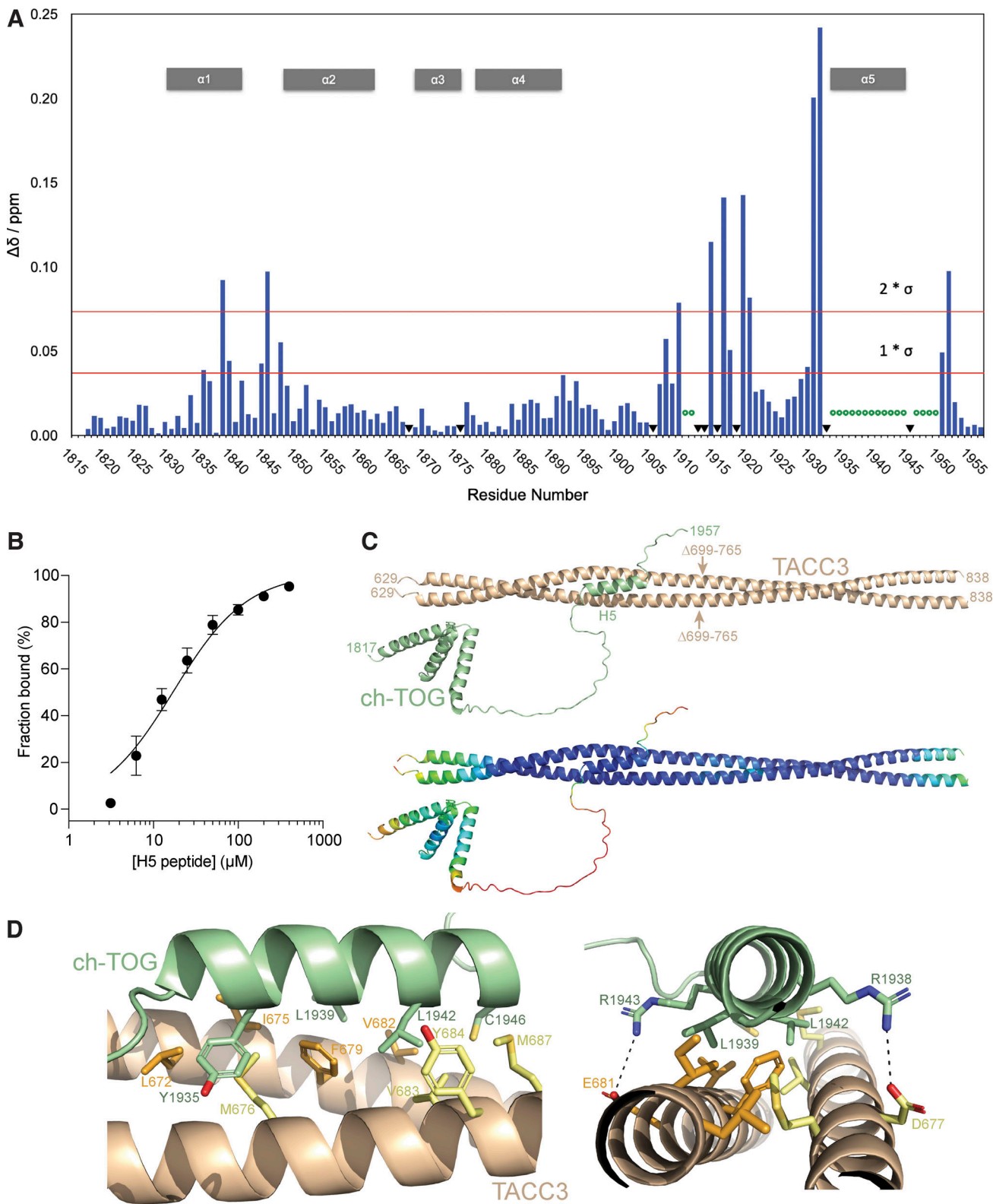

Figure 3. **Interactions of ch-TOG with TACC3. (A)** Chemical shift perturbations observed on the interaction of $^2$H/$^{15}$N labeled ch-TOG 1817–1957 with a threefold excess of TACC3 629–838 Δ699–765. Red lines mark different multiples in standard deviation (as stated) in chemical shift perturbation. Data extracted from the spectra shown in Fig. S3 A. **(B)** MST binding experiment of a synthetic ch-TOG H5 peptide with TACC3 629–838. **(C)** Cartoon representation of an AlphaFold2 Multimer model of the complex between the TACC3 629–838 Δ699–765 dimer (wheat) and ch-TOG 1817–1957 (green). Arrows indicate the position of the deletion in TACC3. The model is shown below colored according to per residue confidence score (pLDDT) in rainbow colors from high (blue) to low (red) confidence. **(D)** Cartoon representations of the AlphaFold2 model showing the interface between the H5 region of ch-TOG (green) and TACC3 (wheat). Sidechains contributing to the interface from TACC3 protomer A (orange), TACC3 protomer B (yellow), and ch-TOG H5 (green) are shown in stick representation.

## Identification of TACC3 Affimers that inhibit the TACC3–ch-TOG interaction

To generate an inhibitor of the TACC3–ch-TOG interaction, we turned to Affimer technology, previously known as Adhiron technology (Tiede et al., 2014, 2017), which can be used to block proteins and their interactions (Haza et al., 2021; Martin et al., 2023; Heseltine et al., 2024). The Affimer scaffold is derived from the consensus sequence of plant phytocystatins, and two structurally adjacent, nine amino acid–loop regions within it have been randomized to generate a library of binder molecules. Biopanning of an Affimer library against the C-terminal TACC domain of TACC3 (residues 629–838 Δ699–765) was carried out to identify antigen-specific clones. The subsequent clones were confirmed by phage ELISA and DNA sequencing, where they were grouped into six unique sequence families. A representative of each family was recombinantly expressed in *E. coli* and purified by affinity chromatography using the C-terminal His-tag on the Affimer protein followed by size-exclusion chromatography for in vitro screening. Of the six Affimers, two precipitated heavily during purification and were not taken forward, while Affimers E4, E5, E7, and E8 were further characterized. In vitro co-precipitation assays using purified recombinant proteins, confirmed the binding of Affimers E4, E7, E8, and E5 to the TACC3 TACC domain (Fig. 4 A). Upon the addition of ch-TOG, the formation of a ternary complex was observed in the case of Affimer E5, but not Affimers E4, E7, and E8, suggesting these bind at a site on the TACC domain that is required for ch-TOG association (Fig. 4 A). In ELISA experiments using purified proteins, Affimers E4, E7, and E8 exhibited clear binding to immobilized TACC3 TACC domain (residues 629–838 Δ699–765) (Fig. 4 B). Affimer E5 displayed a high background, indicating that it is prone to non-specific binding, and was not carried forward.

Next, we identified the binding sites of Affimers E4, E7, and E8 on TACC3 using hydrogen/deuterium exchange mass spectrometry (HDX-MS) (Fig. 4 C and Fig. S3, D–G). The region of TACC3 protected from exchange by the binding of each of these Affimers was residues ~670–682, corresponding to the ch-TOG binding site (Fig. 4 D). This indicates that the observed inhibition of the TACC3–ch-TOG interaction by Affimers E4, E7, and E8 (Fig. 4 A) occurs through their direct occlusion of the ch-TOG binding site on TACC3.

Having confirmed inhibition in vitro, we next wanted to test in human cells the ability of the Affimers to bind TACC3 and to interfere with the TACC3–ch-TOG interaction. We observed colocalization of each of the three Affimers with overexpressed GFP-TACC3 (Fig. S4 B). Proximity ligation assays were performed in HeLa cells to quantify the effects of the Affimers on the formation of complexes of TACC3 and ch-TOG in cells. Transfection with a mCherry-only vector had no effect on TACC3–ch-TOG complex formation, while expression of mCherry-Affimers E4, E7, and E8 resulted in a significant reduction in TACC3–ch-TOG foci, indicating that the Affimers inhibited the interaction in cells, without affecting the expression of ch-TOG or TACC3 (Fig. 4, E–G). Although Affimers have the potential to be used akin to nanobodies (Cordell et al., 2022), none of the Affimers performed well as reagents for

visualization nor for inducible relocalization of endogenous TACC3 to inactivate it (Fig. S4, A–D) (Ryan et al., 2021).

## Disrupting the TACC3–ch-TOG interaction during mitosis

TACC3 and ch-TOG localize to the mitotic spindle in an Aurora-A–dependent manner, where they are part of a multiprotein complex with clathrin and GTSE1 (TACC3–ch-TOG–clathrin–GTSE1) (Ryan et al., 2021). To test the effect of Affimers on subcellular localization of this complex, each Affimer was expressed in GFP–FKBP–TACC3 knock-in HeLa cells, and the distribution of endogenous TACC3 and ch-TOG was compared at metaphase (Fig. 5). As a positive control, cells not expressing Affimers were treated with the Aurora-A kinase inhibitor MLN8237 to abolish TACC3 and ch-TOG spindle localization, as described previously (Booth et al., 2011; Hood et al., 2013) (Fig. 5, A and C). A decrease in ch-TOG spindle localization, comparable with that resulting from MLN8237 treatment was measured for cells expressing Affimers E7 and E8, with a more modest effect for E4 (Fig. 5, B and C). Surprisingly, a small decrease in spindle localization of TACC3 itself was also seen for all three Affimers with respect to untransfected cells (Fig. 5, B and C). However, the reduction was not consistently significant. The decrease in ch-TOG spindle localization could not be attributed to a reduction in microtubule density, which was constant in cells expressing mCherry or mCherry-Affimers (Fig. 5 D). Finally, image averaging showed that ch-TOG remained at the spindle pole but was lost from the spindle to the cytoplasm (Fig. 5 E). In an analogous set of experiments, we confirmed that the localization of endogenous clathrin at the mitotic spindle remained intact in CLTA–FKBP–GFP knock-in HeLa cells expressing TACC3 Affimers (Fig. S4 E). The complete disruption of ch-TOG localization in the presence of Affimers E7 and E8 is consistent with a model where ch-TOG requires an intact interaction with TACC3 for its localization to mitotic spindle microtubules (Hood et al., 2013; Ryan et al., 2021). Moreover, it suggests that the inhibition of the TACC3–ch-TOG interaction by the Affimers is specific and does not interfere with the other interactions within the TACC3–ch-TOG–clathrin–GTSE1 complex.

## A role for the TACC3–ch-TOG interaction at the mitotic centrosome

Next, we sought to ask whether the targeted removal of ch-TOG from mitotic spindle microtubules would produce alterations in spindle morphology. HeLa cells expressing mCherry or mCherry-Affimers were fixed and stained for α-tubulin and pericentrin and imaged in 3D by confocal microscopy. Using a semiautomatic image processing pipeline, we measured several parameters including positioning, tilt, and scaling of the spindle, and found no difference between cells expressing Affimers E7 or E8, compared with Affimer E4 or mCherry alone (Fig. S5 A). However, during this analysis, we noticed that many cells contained more than two distinct pericentrin foci and that these additional foci were associated with small MT asters (Fig. 6 A). We therefore analyzed this using automated image analysis methods. A higher proportion of cells containing more than two pericentrin foci was observed in cells expressing the E7 (30.4%) or E8 (33.3%) Affimer compared with E4 (8.2%) or

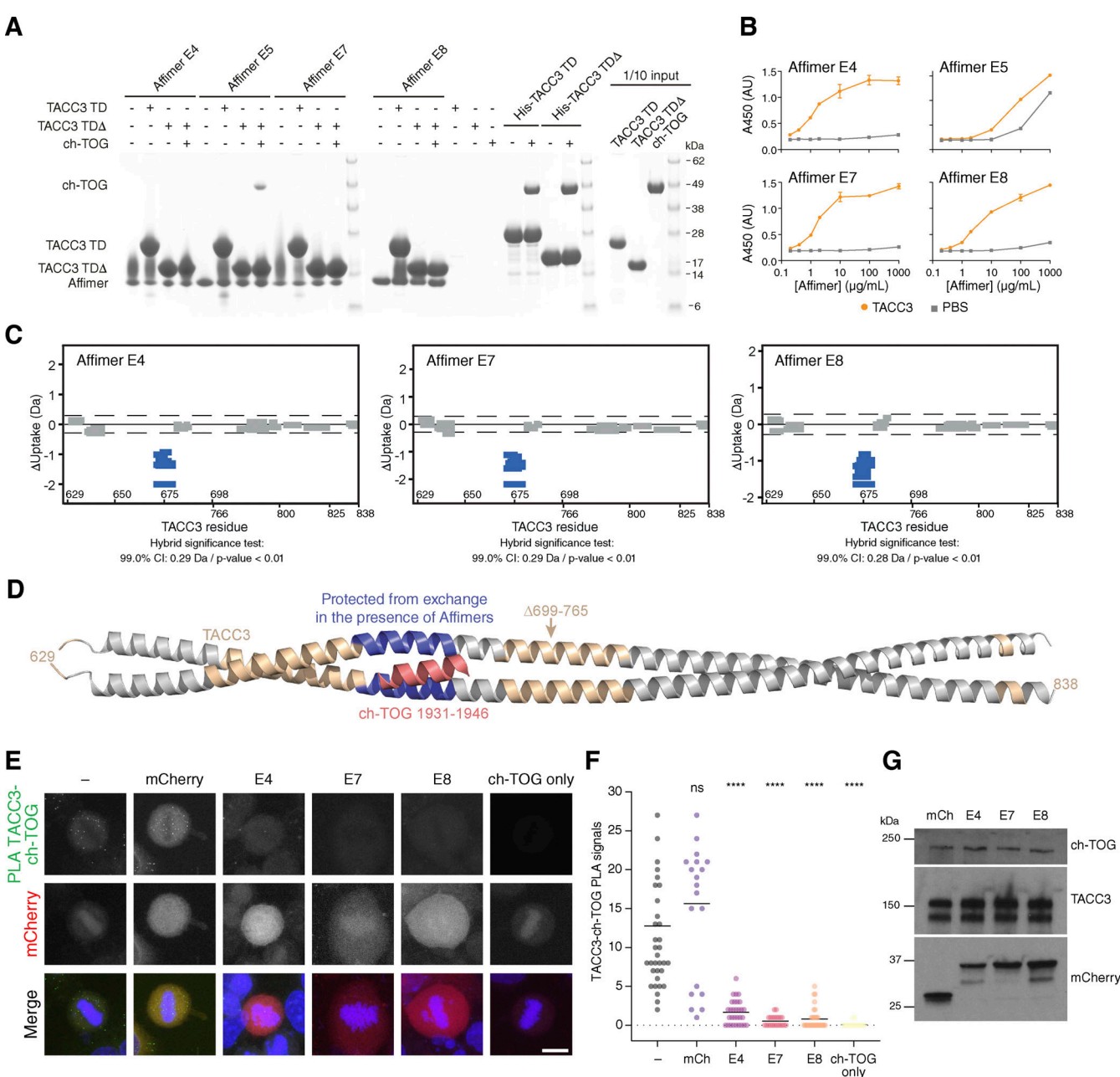

Figure 4. **Isolation of Affimers that bind TACC3 and inhibit TACC3-ch-TOG interaction. (A)** In vitro co-precipitation assay between Affimers, TACC3, and ch-TOG. C-terminal His-tagged Affimers were immobilized on Nickel Sepharose resin and incubated with TACC3 629–838 (TACC3 TD) or TACC3 629–838 Δ699–765 (TACC3 TDΔ). Binding of ch-TOG 1517–1957 in the presence of Affimer was assessed by the addition of ch-TOG to TACC3 TDΔ reactions. **(B)** ELISAs to assess binding between Affimers and TACC3 629–838 Δ699–765. Biotinylated TACC3 629–838 Δ699–765 was immobilized on Streptavidin-coated plates and incubated with an Affimer dilution series (orange circles). Background binding of Affimers to the plate was measured by incubating the proteins in wells coated with PBS (gray squares). Data points are the mean ± standard error of the mean from two experiments. **(C)** Woods plots describing differences in deuterium uptake by residue, after 30 min of exchange, between TACC3 629–838 Δ699–765 in the absence of a binding partner and in the presence of Affimers (as indicated). Woods plots were generated using Deuteros 2.0. Peptides colored in blue are protected from hydrogen/deuterium exchange in the presence of Affimers. Peptides exhibiting no significant difference in exchange between conditions, determined using a 99% confidence interval and a hybrid statistical test (dotted line), are shown in gray. **(D)** Cartoon representation the TACC3 629–838 Δ699–765–ch-TOG complex model with TACC3 colored according to HDX behavior as in C. The region of TACC3 protected from hydrogen/deuterium exchange in the presence of Affimers E4, E7, and E8 is colored blue, and ch-TOG H5 is colored pink. **(E)** Representative confocal micrographs of HeLa cells transfected with mCherry or mCherry-Affimers, as labelled; stained with ch-TOG and TACC3 antibodies for proximity ligation assay. Nuclei are indicated by DAPI staining (blue). Green foci indicate TACC3-ch-TOG protein complexes. Single ch-TOG antibody staining was used as a control for PLA interactions. Scale bars, 10 μm. **(F)** Dot plot graph displaying the number of TACC3-ch-TOG PLA signals per cell from E. Data represent counts from at least 20–30 cells, n = 4. Error bars represent standard deviation of four biological replicates. ****P < 0.0001 in comparison with no transfected sample (–) by one-way ANOVA. **(G)** Western blot to show that ch-TOG and TACC3 levels are not reduced by Affimer expression. Source data are available for this figure: SourceData F4.

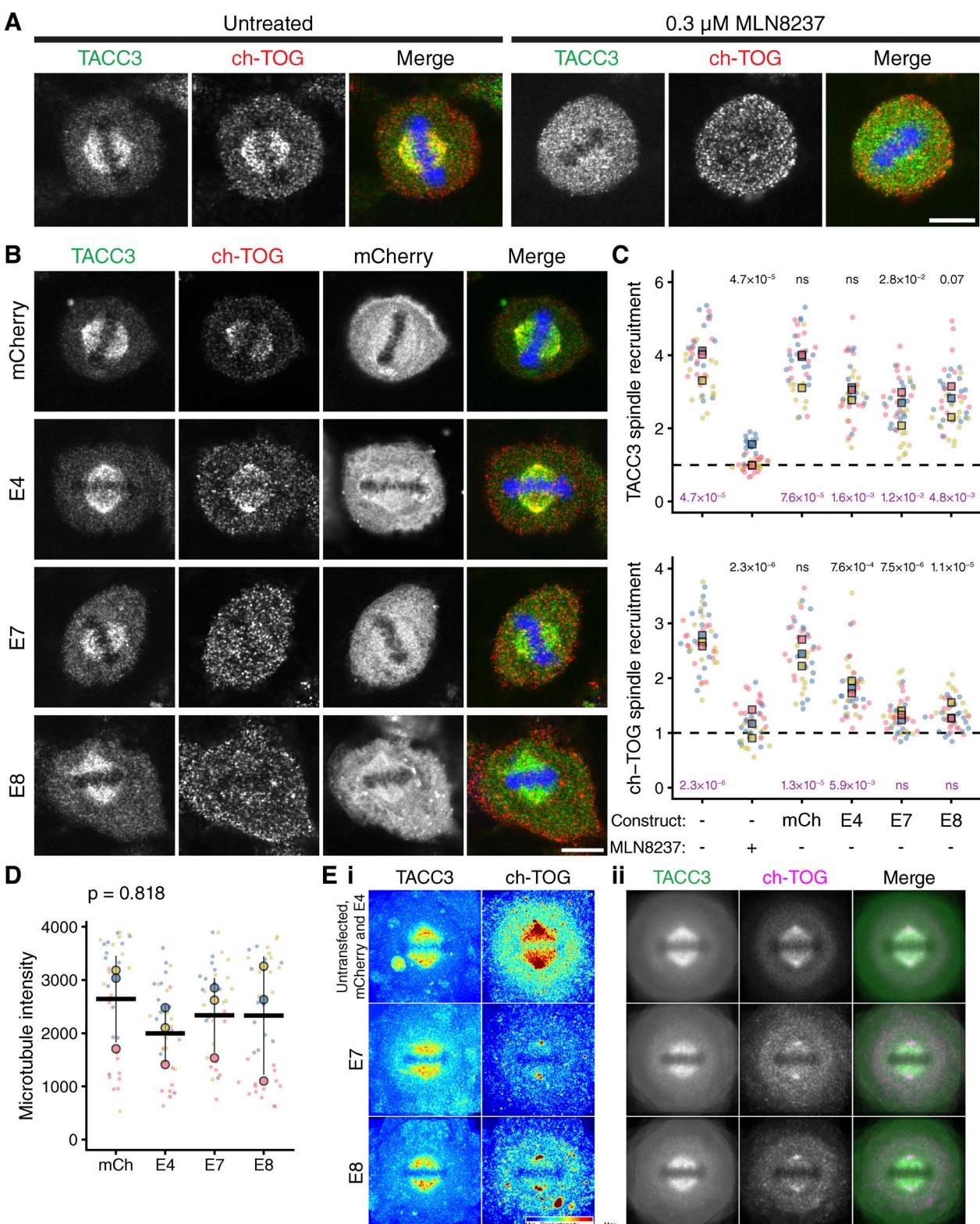

Figure 5. **Affimer-mediated disruption of TACC3–ch-TOG interaction results in lower spindle localization of ch-TOG. (A)** Representative confocal micrographs of untreated or MLN8237-treated (0.3 µM, 40 min) knock-in GFP-FKBP-TACC3 HeLa cells in metaphase. Cells were stained for ch-TOG (red), DNA (blue), and GFP-boost antibody was used to enhance the signal of GFP-FKBP-TACC3 (green). **(B)** Cells expressing mCherry or mCherry-Affimers, labeled as in A. Note, no specific enrichment of Affimers at the spindle (see Fig. S4, A and B). Scale bars, 10 µm. **(C)** Quantification of spindle recruitment of TACC3 and ch-TOG. SuperPlots show single cell measurements as dots, experiment means as outlined markers, colors indicate experiments. Dashed line, no spindle enrichment. Analysis of variance (ANOVA) followed by Tukey's post hoc test is shown above each group, using the untransfected and untreated cells (black) and untransfected MLN8237-treated cells (purple) for comparison. ***, P < 0.001; NS, >0.05. **(D)** Spindle microtubule intensity

was similar in cells expressing mCherry or mCherry-Affimers. SuperPlot of three experiments with mean ± SD represented by the bar and error. **(E)** Averaging images from C shows the relative centrosomal localization of ch-TOG versus TACC3 (i) and the displacement of ch-TOG from the spindle to the cytoplasm in cells expressing Affimers E7 and E8 (ii).

mCherry (2.2%) (Fig. 6 B). Moreover, the additional pericentrin foci appeared smaller in size compared with the two foci that formed the bipolar spindle. Quantification of the total volume of pericentrin foci present in each cell revealed no significant difference in cells expressing Affimer E7 or E8 compared with the mCherry control (Fig. 6 C), suggesting that the additional foci are likely to represent fragments of the PCM rather than amplified centrosomes. Note that this phenotype may be specific to cancer cells because HEK293T cells did not show

fragmentation (two foci in 80% of cells, across conditions). The additional PCNT foci in HeLa cells expressing mCherry-E7 or mCherry-E8 persisted in cells treated with nocodazole, suggesting that they were not caused by spindle microtubule forces (Fig. S5 B). We concluded that, while the TACC3–ch-TOG interaction is not required for normal spindle morphology, it does appear to be required to maintain the structure of the PCM during mitosis. To test this model, HeLa cells expressing mCherry or mCherry-Affimers were fixed and stained to

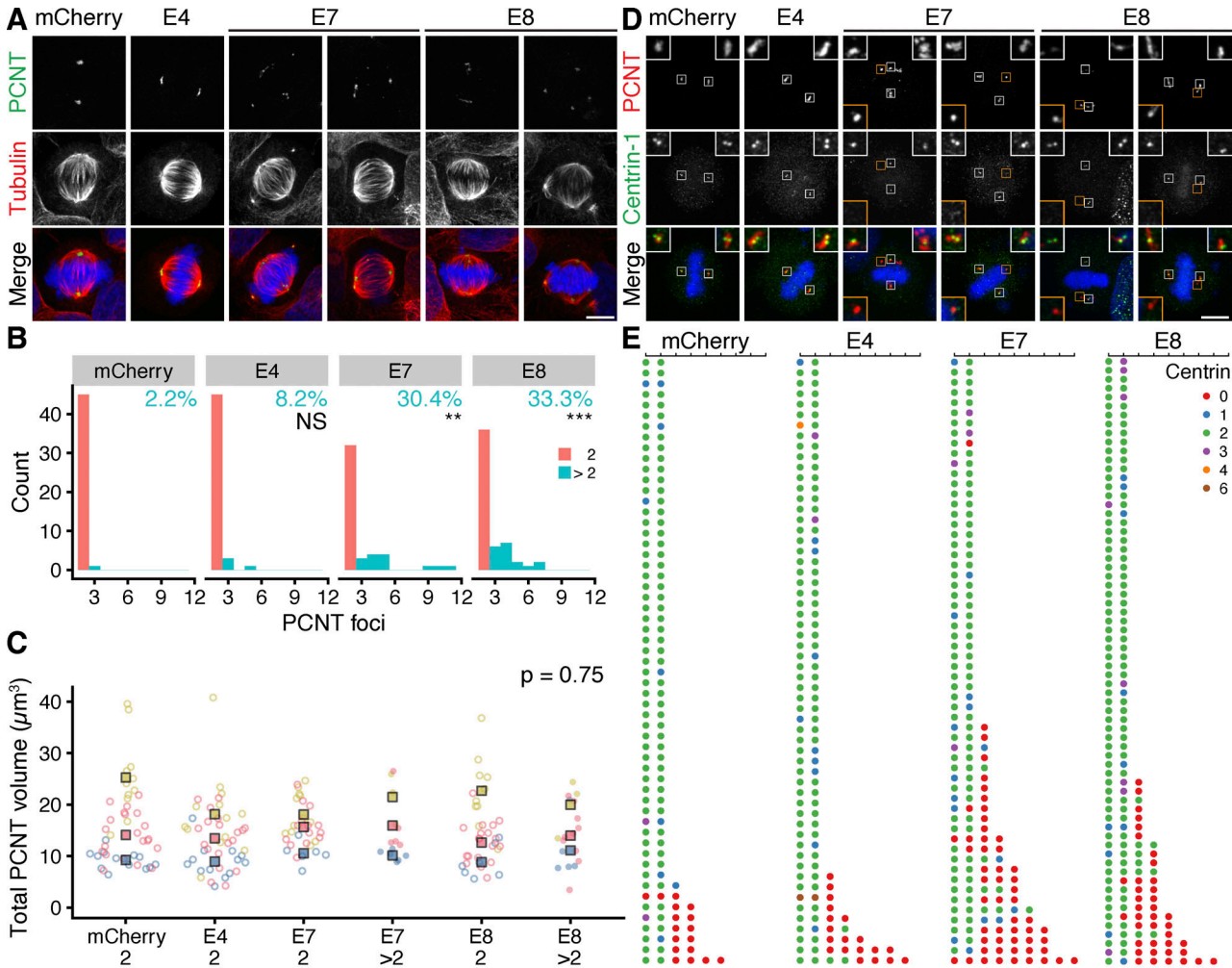

Figure 6.   **Expression of Affimers E7 and E8 leads to fragmentation of pericentrin. (A)** Representative max intensity z projection images of cells expressing mCherry or mCherry-Affimers, as indicated; stained for pericentrin (PCNT, green), α-tubulin (red) and DNA (blue). **(B)** Histograms to show how many cells in each condition had 2 or more PCNT foci. The percentage of cells with >2 foci is indicated. Fisher's exact test was used to test for association between the type of Affimer expressed and the PCM foci category. Bonferroni adjustment was used to calculate P values. **(C)** Superplots to show the total volume of pericentrin foci. Dots, single cells; markers, mean of each experiment, colors indicate experiments. In B and C, cells with exactly two pericentrin foci are shown in salmon/ filled dots and those with >2, turquoise/empty dots; P value from two-way ANOVA between Affimer conditions. **(D)** Representative max intensity z projection images of cells expressing mCherry or mCherry-Affimers, as indicated; stained for pericentrin (PCNT, red), centrin-1 (green) and DNA (blue). Insets show a 2.5 × zoom of the PCM, orange insets show sites of fragmentation. Scale bars, 10 μm. **(E)** Quantification of the number of pericentrin foci and the number of centrin-1 foci associated with each PCNT focus. Each dot represents a PCNT focus, with the color of the dot indicating the number of centrin-1 foci present, as described in the legend; each row is a different cell. Data is shown for n = 57–68 cells per condition over three independent experiments.

visualize pericentrin and centrin1 as markers for the PCM and centrioles, respectively (Fig. 6 D). This analysis revealed that in the majority of cells containing >2 pericentrin foci, centrin-1 is absent from the additional sites (Fig. 6 E), confirming that they are detached fragments of PCM.

To address the possibility that Affimer E7- and E8-mediated PCM fragmentation might arise due to an off-target effect, we interfered with the TACC3–ch-TOG interaction with an alternative approach. In cells depleted of endogenous ch-TOG, we expressed RNAi-resistant ch-TOG-GFP, a ch-TOG-GFP L1939A, L1942A mutant that cannot bind TACC3, or GFP as a control (Gutiérrez-Caballero et al., 2015). As expected, in ch-TOG–depleted cells expressing GFP, ~50% of mitotic cells contained multipolar spindles (Fig. 7, A and B). This is consistent with previous reports of ch-TOG depletion, highlighting its role in spindle pole organization (Gergely et al., 2003; Cassimeris and Morabito, 2004). Expression of full-length ch-TOG rescued the multipolar phenotype, with only 14.2% of cells containing >2 pericentrin foci (Fig. 7, A and B). In contrast, expression of the ch-TOG L1939A, L1942A mutant was associated with a pericentrin fragmentation phenotype in 36% of cells (Fig. 7, A and B). These results were similar to those observed in the presence of Affimers E7 and E8 (Fig. 6), and again the total volume of pericentrin in cells with fragmentation was similar to that in cells expressing full-length ch-TOG (Fig. 7 C). Taken together, these data suggest that the observed PCM fragmentation phenotype is a consequence of blocking the TACC3–ch-TOG interaction.

## TACC3–ch-TOG interaction is required for maintenance of centrosomal integrity during mitosis

From experiments in fixed cells, it was unclear whether fragmentation occurs prior to the cell entering mitosis, during spindle assembly, or once the spindle has formed. To clarify this point, we used live-cell imaging to visualize the PCM (mEmerald-γ-tubulin) and DNA (SiR-DNA) in cells expressing mCherry-Affimers, or mCherry as a control (Fig. 8 A). Consistent with our previous experiments, the proportion of cells with >2 γ-tubulin foci during metaphase was higher in Affimer E7- (20.7%) and Affimer E8 (19.4%)-expressing cells, compared with mCherry (6.7%) and Affimer E4 (11.1%). Interestingly, in cells expressing Affimers E7 and E8, the metaphase–anaphase transition was when the largest fraction of cells acquired supernumerary γ-tubulin foci, with 13.4% and 15.3% of cells displaying this phenotype, respectively, by this stage (Fig. 8 B). In these cells, the γ-tubulin foci underwent fragmentation to form smaller foci that remained close to the spindle before the cell divided (Fig. 8 A). Moreover, a prolonged metaphase–anaphase transition was observed in cells with this phenotype. Control cells maintained two distinct γ-tubulin foci throughout the observed stages and divided without a delay (Fig. 8 A). There was no significant change in overall mitotic progression in cells expressing Affimers E7 or Affimer E8 compared with Affimer E4 or mCherry alone (Fig. 8 C). However, when the mitotic timings of Affimer E7- and Affimer E8-expressing cells containing 2 or >2 γ-tubulin foci during metaphase were compared (Fig. 8 D), a delay in metaphase–anaphase progression was revealed: cells that exhibited PCM fragmentation during metaphase had a median metaphase–anaphase timing of 60 min (E7) and 102 min (E8),

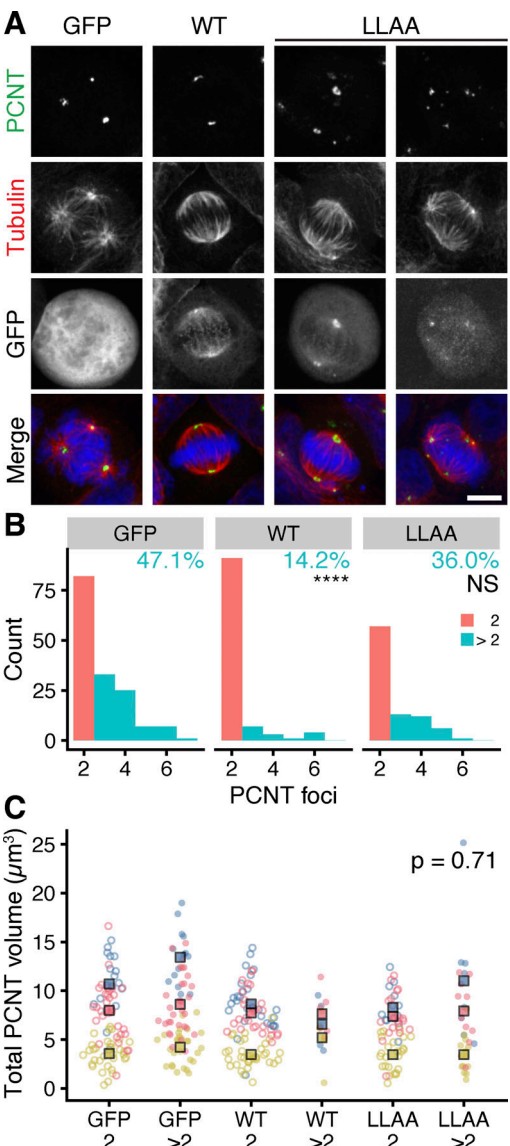

Figure 7. **Expression of a ch-TOG mutant deficient in binding TACC3 results in fragmentation of pericentrin in mitotic HeLa cells. (A)** Representative max intensity z projection images of HeLa cells co-expressing shRNA against ch-TOG, and either GFP, RNAi-resistant ch-TOG-GFP (WT) or ch-TOG(L1939,1942A)-GFP (LLAA). Cells were stained for pericentrin (green), α-tubulin (red), and DNA (blue). Scale bar, 10 μm. **(B)** Histograms to show how many cells in each condition had two or more PCNT foci. The percentage of cells with >2 foci is indicated. Fisher's exact test was used to test for association between the protein expressed and the PCM foci category. Bonferroni adjustment was used to calculate P values. **(C)** Superplots show the total volume of pericentrin foci. Dots, single cells; markers, mean of each experiment, colors indicate experiments. In B and C, cells with exactly two pericentrin foc are shown in salmon/filled dots and those with >2, turquoise/empty dots; P value from two-way ANOVA, between expression conditions.

while those that cells that contained two foci throughout metaphase had a median metaphase–anaphase timing of 30 min. Taken together, these data show that inhibition of the TACC3-ch-TOG interaction causes fragmentation of the PCM during metaphase, which is accompanied by a delay in the transition to anaphase.

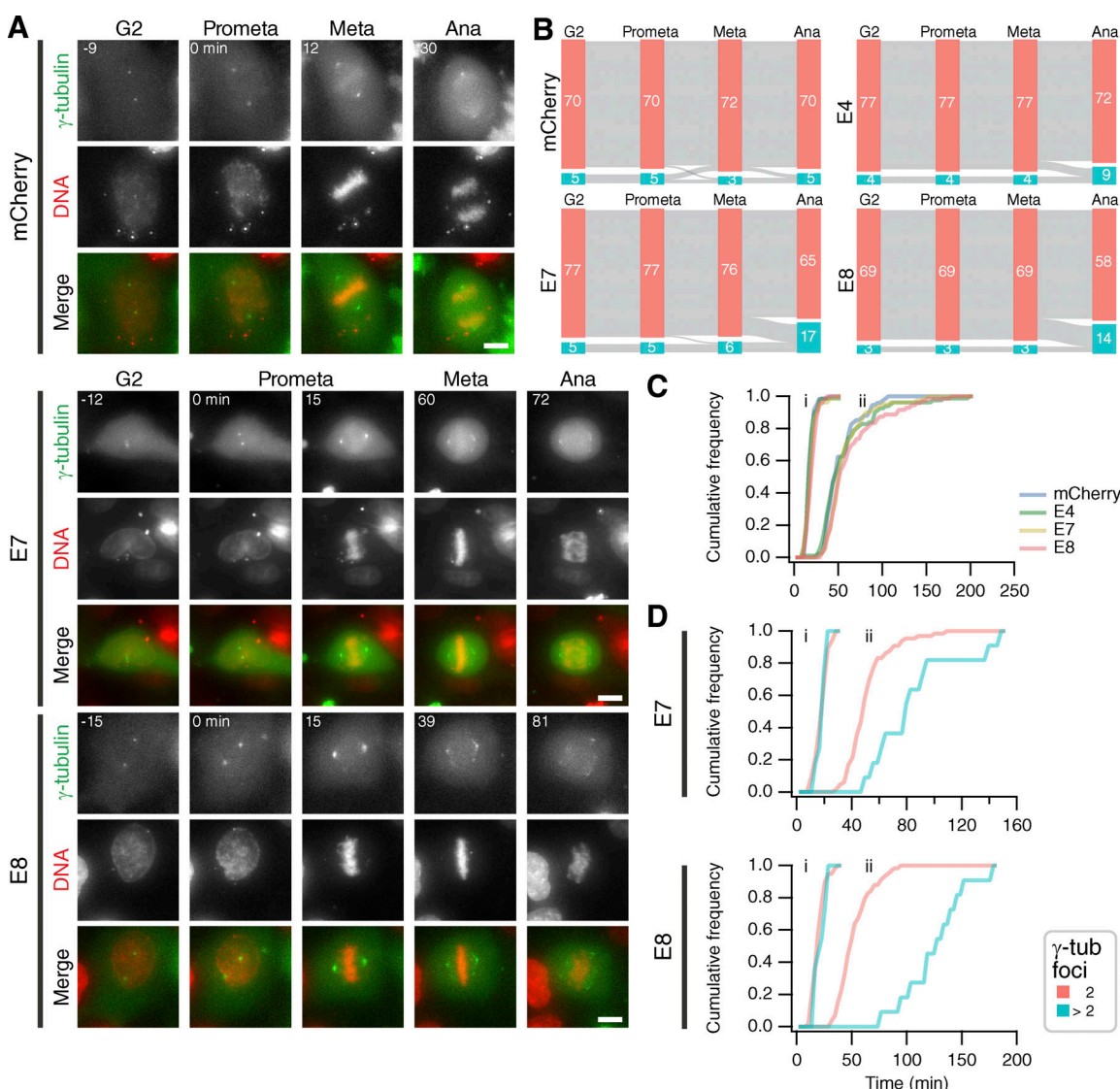

Figure 8. **Blocking TACC3–ch-TOG interaction with Affimers results in fragmentation of PCM and mitotic delay. (A)** Stills from live cell imaging experiments to track the number of γ-tubulin foci in cells expressing mEmerald-γ-tubulin (green) and the indicated mCherry or mCherry-Affimers constructs (not shown), SiR-DNA staining is shown (red). Scale bar, 10 μm. **(B)** Sankey diagrams to show the number cells containing supernumerary γ-tubulin foci at the indicated stages of cell division. The number of γ-tubulin foci was tracked from G2-prometaphase, prometaphase-metaphase and metaphase-anaphase. Numbers in each node represent the number of cells observed at each stage, as labeled. Node color represents the number of γ-tubulin foci in the cell, those with two are shown in salmon and those >2, turquoise. Data is pooled from four independent overnight experiments. **(C)** Mitotic progression of HeLa cells expressing mCherry or mCherry-Affimers. Cumulative histograms of prometaphase to metaphase **(i)** and prometaphase to anaphase **(ii)** timings. Number of cells analyzed: mCherry, 75; Affimer E4, 81; Affimer E7, 82; Affimer E8, 72. **(D)** Frequencies of Affimer E7- or Affimer E8-expressing cells shown in A, comparing timings of cells with two γ-tubulin foci (2; salmon) during metaphase with cells that undergo PCM fragmentation during metaphase (>2; turquoise). Number of cells: (2 and >2 foci, respectively): Affimer E7, 65 and 11; Affimer E8, 58 and 11.

## Discussion

In this paper, we described the structural details of the interaction between ch-TOG and TACC3. We isolated Affimers that bound TACC3 at the site of interaction with ch-TOG and displaced it. These new molecular tools could be expressed in cells and the function of the TACC3–ch-TOG interaction dissected. We uncovered a role for this interaction in stabilizing the pericentriolar matrix during mitosis.

The TACC3–ch-TOG interaction is through the H5 of ch-TOG binding to the parallel dimeric coiled-coil of TACC3 to form a short trimeric coiled-coil domain. In the absence of this interaction, H5 packs loosely against the four-helix core of this short domain, and our NMR measurements showed the dynamicity of the interchange between these two states. The interior of the core region is entirely filled with hydrophobic amino acids, and essentially all of them are highly conserved in the sequence of this domain from insects to mammals. This suggests that the core domain is a common feature of XMAP215/ch-TOG family proteins. The contacts made between H5 and the core domain are not strong, but they are sufficient to hold H5 in position and keep it in a helical conformation. Interestingly, in this position, the two leucines (Leu1939 and Leu1942) that are

important for binding to the TACC domain of TACC3 (Gutiérrez-Caballero et al., 2015) are pointing inwards and are not accessible from the outside. Therefore, to bind TACC3, H5 has to be dislodged from the core domain to make these leucines accessible. Such cryptic binding interfaces are seen in other proteins where a range of mechanisms are employed for their release (Gingras et al., 2006). How H5 becomes dislodged from the core to bind TACC3 and what triggers this event is an interesting question for future investigation.

The TACC3 Affimers developed in this study were sufficient to disrupt the TACC3–ch-TOG interaction in vitro and in cells. The specific targeting of a single protein–protein interaction is very useful, especially in the case of TACC3–ch-TOG where the proteins participate in other complexes, alone and in combination as well as likely functioning individually. Small molecule destabilization of TACC3, TACC3 depletion by RNAi, TACC3 knocksideways, or Aurora-A inhibition all remove the entire TACC3–ch-TOG–clathrin–GTSE1 complex from the mitotic spindle as well as interfere with TACC3 functions at the spindle pole (LeRoy et al., 2007; Wurdak et al., 2010; Booth et al., 2011; Cheeseman et al., 2013; Hood et al., 2013; Akbulut et al., 2020). Similarly, the depletion of ch-TOG has several other effects besides targeting the TACC3–ch-TOG interplay (Gergely et al., 2003; Herman et al., 2020). Targeting the interaction using Affimers means that TACC3 and clathrin stay in place on the mitotic spindle microtubules and the structure of the spindle is unaffected but that ch-TOG specifically is absent. While similar results can be achieved by specific mutation of the binding site, the advantage of using Affimers for disruption is that they have the potential to be deployed to cells without the need for introducing such mutations.

Interrupting the TACC3–ch-TOG interaction during mitosis in living cells resulted in a spindle pole defect: PCM fragmentation. TACC3 and ch-TOG have previously been shown to be involved in centrosome clustering in cancer cells, either via interaction with integrin-linked kinase or with KIFC1 (Fielding et al., 2011; Saatci et al., 2023). However, the phenotype we uncovered is distinct since we saw multiple pericentrin foci in mitotic cells expressing the inhibitory Affimers, of which only two contained centrioles. Instead, the phenotype points to a recently described property of TACC3 at spindle poles. First, in *Drosophila* spindles, the PCM is templated by a cnn-containing structure with a more liquid-like d-TACC region associated with it (Wong et al., 2025). Second, in *C. elegans* oocyte meiotic spindles, XMAP215/ZYG-9 and TACC/TAC-1 act at multiple times during assembly to promote spindle pole integrity and stability (Harvey et al., 2023). Our data suggest that the TACC3–ch-TOG interaction is important for maintaining the PCM around the centrosomes and that fragmentation results in a mitotic delay. Precisely how the interaction does so requires further investigation. Nonetheless, the discovery of a novel mechanism involving TACC3–ch-TOG that maintains PCM integrity during mitosis underscores the importance of using precise tools such as Affimers to investigate protein–protein interactions.

# Materials and methods
## Molecular biology
The following plasmids were generated in the course of previous work: mNeonGreen-EB3, pMito-mCherry-FRBK70N, GFP-TACC3, pBrain-GFP-shch-TOG, pBrain-ch-TOGKDP-GFP-shch-TOG, and pBrain-ch-TOGDPGFP(LL1939,1942A)-shch-TOG (Gutiérrez-Caballero et al., 2015); pETM6T1 TACC3 629–838 and pETM6T1 ch-TOG 1517–1957 (Hood et al., 2013); and mEmerald-γ-tubulin was from Addgene #54105.

ch-TOG 1467–2032 was made by amplifying the corresponding cDNA region using the following primers, digestion of the TOPO-cloned PCR product using NcoI and MfeI, and ligation into pETM6T1 cut with NcoI and EcoRI: chTOG_1467_NcoI_F (5′-CCATGGCCCGAAGCATGAGTGGGCATCCTGAGGCAGCCCAGATGG-3′), chTOG_2032_MfeI_R (5′-CAATTGTCATTTGCGACTGCTCTTTATTCTCTCCAGTCTTTTTTTCAAGTCGTC-3′). ch-TOG truncates were produced by amplifying the corresponding cDNA regions from ch-TOG 1467–2032 using the following primers, digestion of the PCR products with NcoI and XhoI, and ligation into pETM6T1 cut with the same restriction enzymes: chT1517start (5′-CGCGCCATGGTCCTTATTCCTGAACCCAAGATC-3′), chT1817startNco (5′-CGCGCCCATGGCATCTCGAATAGATGAAAAATCATCAAAGGC-3′), chT1827startNco (5′-CGCGCCATGGCCAAAGTGAATGATTTCTTAGCTGAG-3′), chT1804Xhoend (5′-CGCGCTCGAGTCACTGGTCCATACTGTGCTTC-3′), chT1815Xhoend (5′-CGCGCTCGAGTCACTTTTCTGTTTCCTTATCAGACTTGC-3′), chT1957Xhoend (5′-CGCGCTCGAGTCAAGGTCGGTCATCTTGC-3′).

pETM6T1 TACC3 629–838 Δ699–765 was produced by deletion QuikChange site-directed mutagenesis (Agilent) of the TACC3 629–838 construct using primers: T3d699/765F (5′-GCAGAAGGAACTTTCCAAAGCCCTGAAGGCCCACGCG-3′), T3d699/765R (5′-CGCGTGGGCCTTCAGGGCTTTGGAAAGTTCCTTCTGC-3′). For generation of the TACC3 629–838 Δ699–765-Avi expression construct, the following sequence was digested with SpeI and SalI and cloned into pETM6T1 digested with SpeI and XhoI to generate a vector providing a C-terminal Avi-tag: 5′-ACTAGTGGTTCTGGTCATCATCACCACCATCACGATTACGATATCCCAACGACCGAAAAC-TTGTATTTCCAGGGCGCCATGGTCGGATCCGAAGAATTCGCGCGCGCGGCCGCAAAGCTT-CTCGAGGGTCTTAACGATATTTTTGAAGCTCAGAAAATTGAATGGCACGAGGCATGAGTC-GAC-3′. TACC3 629–838 Δ 699–765 was amplified with the following primers: tacc3_629_NcoI_F (5′-AAAGTTACCATGGTCTCCACCGGACCTATAGTGGACCTGCTCCAGTAC-3′), tacc3_838_XhoI_R (5′-ATCATATCTCGAGGATCTTCTCCATCTTGGAGATGAGGTCGTCGCAGAT-3′), and the product was digested with NcoI and XhoI for ligation into the modified pETM6T1-Avi-tag plasmid cut with the same enzymes to create a construct with a N-terminal TEV-cleavable His-NusA tag and a C-terminal Avi-tag.

TACC3 and ch-TOG point mutations were introduced by QuikChange site-directed mutagenesis (Agilent) using the following primers: Tcys662aF (5′-GGAGCAGGGCTGAGGAGCTCCACGGGAAGAAC-3′), Tcys662aR (5′-GTTCTTCCCGTGGAGCTCCTCAGCCCTGCTCC-3′), Tcys749aF (5′-GAACGAAGAGTCACTGAAGAAGGCCGTGGAGGATTACC-3′), Tcys749aR (5′-GGTAATCCTCCACGGCCTTCTTCAGTGACTCTTCGTTC-3′), Tcys828aF

(5′-ACGAGGAGCTGACCAGGATCGCCGACGACCTCATCTC-3′), Tcys828aR (5′-GAGATGAGGTCGTCGGCGATCCTGGTCAGCTCC TCGT-3′), chTY1935AFwd (5′-GCCATCTGTCGCCTTGGAAAG GCTAAAG-3′), chTY1935ARv (5′-CTTTAGCCTTTCCAAGGCGAC AGATGGC-3′), chTR1938AFwd (5′-CTACTTGGAAGCGCTAAA GATCCTCCG-3′), chTR1938Rv (5′-CGGAGGATCTTTAGCGCT TCCAAGTAG-3′), chTL1939RFwd (5′-CTACTTGGAAAGGCGAAA GATCCTCCG-3′), chTL1939RRv (5′-CGGAGGATCTTTCGCCTT TCCAAGTAG-3′), chTK1940AFwd (5′-GGAAAGGCTAGCGAT CCTCCGACAG-3′), chTK1940ARv (5′-CTGTCGGAGGATCGCTAG CCTTTCC-3′), chTL1942RFwd (5′-GGCTAAAGATCCGCCGAC AGCGATG-3′), chTL1942RRv (5′-CATCGCTGTCGGCGGATCTTT AGCC-3′), chTR1943AFwd (5′-GGCTAAAGATCCTCGCACAGC GATGTGG-3′), chTR1943ARv (5′-CCACATCGCTGTGCGAGGATC TTTAGCC-3′), chTR1945AFwd (5′-CCTCCGACAGGCATGTGG TCTGG-3′), chTR1945ARv (5′-CCAGACCACATGCCTGTCGGA GG-3′), chTL1948RFwd (5′-CGATGTGGTCGGGACAACACAAAG CAAG-3′), chTL1948RRv (5′-CTTGCTTTGTGTTGTCCCGACCAC ATCG-3′).

Selected Affimer cDNA sequences were subcloned from the phagemid display vector, pBSTG1, into pET11a (Tiede et al., 2014). Sequences encoding TACC3-targeted Affimers E4, E7, and E8 were cloned into pmCherry-C1 (cut and paste) and pmCherry-N1 (PCR, cut and paste) vectors to give mCherry-Affimer or Affimer-mCherry constructs, respectively. The mCherry-Affimer orientation was found to be the most effective. Primers used for insertion into pmCherry-N1: JS034 (5′-TAAGCAAGCGCTGCCACCATGGCTAGCAACTCCCTGGAA ATC-3′) and JS035 (5′-TGCTTAGTCGACGCAGCGTCACCAACC GGTTTG-3′). To make FKBP-GFP-Affimer, the following primers were used for insertion of each Affimer sequence into pFKBP-GFP-C1: JS043 (5′-TGCTTAGAGCTCTTATGCAGCGTC ACCAACC-3′) and JS044 (5′-TAAGCATCTAGAGCCACCATG GCTAGCAACTCCCTGGAAATC-3′).

### Protein expression and purification
All proteins were expressed and purified as described in an earlier work (Hood et al., 2013). In brief, *E. coli* BL21(DE3)RIL cells were transformed with the selected plasmid and grown in LB media supplemented with the appropriate antibiotics at 37°C until an $OD_{600}$ ~0.6 was reached. Recombinant protein expression was induced by the addition of 0.6 mM IPTG followed by overnight incubation at 21°C. To produce selectively biotinylated TACC3 629–838 Δ699–765-Avi protein, *E. coli* B834(DE3) cells were cotransformed with pETM6T1 TACC3 629–838 Δ699–765-Avi and pBirAcm (Avidity), encoding birA biotin ligase, and grown as normal but with the addition of biotin to a final concentration of 50 μM upon induction with IPTG. Cell pellets were resuspended in 50 mM Tris pH 7.5, 0.3 M NaCl, and lysed by sonication. Lysates were clarified by centrifugation and applied to a 5 ml HiTrap Chelating Sepharose column (Cytiva). Non-bound proteins were washed from the column followed by elution of His-tagged proteins with a linear gradient of imidazole in 50 mM Tris pH 7.5, 0.3 M NaCl. Directly after affinity chromatography, Affimers and His-NusA proteins were subjected to size-exclusion chromatography on a HiLoad 16/600 Superdex 200 pg column (Cytiva) into 20 mM Tris pH 7.0, 50 mM NaCl,

5 mM 2-mercaptoethanol. Untagged proteins were generated by overnight cleavage with TEV protease followed by reverse affinity and anion-exchange chromatography, to remove the His-NusA tag, and size-exclusion chromatography into 20 mM Tris pH 7.0, 50 mM NaCl, and 5 mM 2-mercaptoethanol.

### Co-precipitation assays
Co-precipitation assays were performed with recombinant, purified proteins as described previously (Hood et al., 2013). In summary, 100 μg His-tagged protein was immobilized on 20 μl Nickel Sepharose resin (Cytiva) equilibrated in 50 mM Tris pH 7.5, 150 mM NaCl, 40 mM imidazole, and 0.1% Tween 20 for 2 h with rotation at 4°C. Beads were washed three times with 1 ml of reaction buffer after which 100 μg of interaction partner(s) was added and incubated with the beads for a further 2 h. The beads were washed three times with 1 ml of reaction buffer and resuspended in 20 μl SDS-loading buffer prior to analysis by SDS-PAGE.

### Analytical ultracentrifugation
Sedimentation velocity experiments were performed at 60 k rpm in a Beckman XL-I analytical ultracentrifuge using the An-60 Ti rotor and Spin Analytical 2-sector cells. Samples of TACC3 629–838 were run at 2, 0.75, and 0.2 mg/ml in 50 mM Tris pH 7.5, 150 mM NaCl after extensive temperature equilibration at 4°C, and data were collected in both absorbance and interferences modes. The vbar of the protein, and the density and viscosity of the buffer at 4°C were calculated using the program SEDNTERP (Philo, 2023). Sedimentation velocity traces were fitted with the $c(s)$ with one discrete component model in SEDFIT. Parameters for the discrete component were fixed at their default values or left floating. The confidence level of maximum entropy regularization was varied between 0.68 and 0.95 to give essentially identical results.

### Electron paramagnetic resonance
MTSL-labeled TACC3 proteins were produced as described in previous work (Concilio et al., 2016). In brief, purified TACC3 proteins were desalted on a HiPrep 26/10 desalting column (Cytiva) into EPR buffer (20 mM Tris pH 7.0, 200 mM NaCl, 5 mM $MgCl_2$, 10% glycerol), and a 10-fold molar excess of MTSL was added and the reaction was incubated at 4°C with rotation overnight. After the labeling reaction, excess MTSL and any aggregated protein were removed from the sample by size-exclusion chromatography using a HiLoad 16/600 Superdex 200 pg column (Cytiva) equilibrated in EPR buffer.

Continuous-wave EPR (CW EPR) was measured to ascertain the possible presence of dipolar broadening. CW EPR spectra were recorded 120 K on a Bruker Micro EMX spectrometer using a super-high sensitivity probe head at 9.4 GHz using a microwave power of 20 mW and modulation amplitude of 1 G. Pulsed electron–electron double resonance (PELDOR or DEER) spectroscopy measurements separate dipole–dipole coupling between spins, which is inversely proportional to the cube of their distance (Milov et al., 1981, 1984). It can measure distances between spin labels on the nanometer scale, typically between 1.5 and 6 nm (Jeschke, 2012). TACC3 629–838 C749A, C828A and

TACC3 629–838 C662A, C749A (50 µM) containing 30% glycerol were used for the studies. DEER measurements were performed at 50 K on a Bruker Elexsys 580 spectrometer. The four-pulse DEER sequence $\pi/2_{\nu obs}-\tau_1-\pi_{\nu obs}-t-\pi_{\nu pump}-(\tau_1+\tau_2-t)-\pi_{\nu obs}-\tau_2-echo$ was applied (Pannier et al., 2000), with $\pi/2_{\nu obs}$ pulse length of 16 ns, $\pi_{\nu obs}$ pulse length of 32 ns, and $\pi_{\nu pump}$ pulse length of 32 ns. Pump pulses were applied at the maximum of the field sweep spectrum with the observed pulses lower than 65 MHz. Phase cycling was applied. The software DEERAnalysis2022 (Jeschke et al., 2006) was used to subtract the exponential background decay due to intermolecular interactions and to calculate the interspin distance distribution by Tikhonov regularization.

### NMR spectroscopy

Protein expression, purification, and NMR data collection were carried out as described previously (Rostkova et al., 2018). ch-TOG fragments cloned in pETM6T1 were expressed in *E. coli* BL21 cells overnight at a temperature of 18°C. Cells were lysed by freeze-thawing with the addition of lysozyme and DNAseI. Proteins were purified on an AKTApure chromatography system using 5 ml HisTrap Ni-NTA columns. The fusion protein was cleaved off with TEV protease upon which the chromatography was repeated (after removing imidazole by dialysis). The purified protein was pooled and concentrated in VivaSpin 20 concentrators with 3 kDa MWCO. NMR samples were prepared in a buffer of 20 mM HEPES, 150 mM glutamic acid/arginine, and 2 mM DTT pH 7.2 with protein concentrations between 200 and 500 µM. The assignment data (BMRB entry: 27235) yielded chemical shifts that were used by DANGLE (Cheung et al., 2010) to calculate backbone dihedral angle constraints. Distance constraints were extracted from 3D $^{15}$N and $^{13}$C resolved NOESY-HSQC (standard pulse sequence provided by the manufacturer, water suppression using watergate) spectra recorded at 800 MHz on a Bruker Neo spectrometer equipped with a TCI cryoprobe. Residual dipolar couplings were extracted from an In-Phase/Antiphase HSQC experiment (Ottiger et al., 1998) recorded in the presence of 5 mg/ml of Pf1 phage (ASLA biotech). All NMR spectra processing was done in Topspin 3.4 (Bruker). All NMR data analysis and preparation for structure calculation were done using CCPNMR analysis 2.4 (Vranken et al., 2005; Skinner et al., 2015). In short, 3D NOESY peaks were picked manually according to the established assignment. Structure calculation was initiated with ARIA (Rieping et al., 2007) using the standard, default protocol. All NOESY distance constraints were initially unassigned allowing for the ARIA protocol to assign all distance constraints. Dihedral constraints were introduced in the third round of structure calculation. Hydrogen bond constraints were introduced at the same time for hydrogen bonds that were observed in at least 15 of the final 20 structures at the end of the previous calculation. Once the vast majority of distance constraints were assigned or an ambiguous assignment was confirmed after manual inspection, structure calculation was completed in XPLOR-NIH (Schwieters et al., 2003) using default protocols provided with the software at which point also the RDC constraints were introduced.

### Microscale thermophoresis (MST)

The TACC domain (residues 629–838) and the synthetic H5 peptide (residues 1929–1947) were transferred into a measurement buffer consisting of 20 mM sodium phosphate, 50 mM NaCl, 2 mM DTT, 0.05% Tween, and 0.02% NaN₃. The TACC3 domain was fluorescently labeled via NHS coupling using the standard NHS Red labeling kit following the manufacturer's protocol (Nanotemper). MST experiments were recorded on a Nanotemper Monolith instrument using a concentration of the fluorescently labeled TACC domain of 600 nM while the concentration of the H5 peptide was varied in the range of 0–400 µM. Samples were measured using premium capillaries (Nanotemper). For the creation of the binding curve fluorescence intensity, ratios were calculated from the reference reading prior to heating the sample to the end of the thermophoresis curve. Extracted fluorescence ratios were exported and analyzed in Prism 10 (Graphpad). The binding curve was fitted to a one-site–specific binding model by nonlinear regression to obtain the affinity measurement. The experiment was performed three times and the standard deviation of the three repeats was taken as the experimental error.

### Structural modeling

To generate models of TACC3, a TACC domain in association with ch-TOG, the AlphaFold2 neural network (Jumper et al., 2021) via the ColabFold pipeline (Mirdita et al., 2022) was used. The input proteins were based on fragments that interact biochemically: human TACC3 629–838 Δ699–765 (two copies) and human ch-TOG 1817–1957. ColabFold was executed using default settings: multiple sequence alignment with MMseqs2, no use of templates nor Amber, and five models generated. Models were visualized using PyMol.

### Isolation of TACC3 Affimers

Selection of anti-TACC3 Affimers by phage display was performed as described previously (Tiede et al., 2014, 2017). In brief, 1 µg of biotinylated TACC3 629–838 Δ699–765-Avi was bound to streptavidin-coated wells (Pierce), washed, and then 1 × 10$^{12}$ cfu phage was added for 2 h with shaking. Panning wells were washed nine times and phage-eluted with 100 µl 50 mM glycine–HCl (pH 2.2) for 10 min, neutralized with 15 µl 1 M Tris–HCL (pH 9.1), further eluted with 100 µl triethylamine 100 mM for 6 min, and neutralized with 50 µl 1 M Tris–HCl (pH 7). The eluted phage was used to infect 5 ml of ER2738 cells for 1 h at 37°C and 90 rpm and then plated onto LB agar plates with 100 µg ml$^{-1}$ carbenicillin and grown overnight. Colonies were scraped into 5 ml of 2TY medium, inoculated in 8 ml of 2TY medium with carbenicillin, and infected with ~1 × 10$^9$ M13K07 helper phage at OD600 of 0.5–1.0. After 1 h at 90 rpm, kanamycin was added to 25 µg ml$^{-1}$ overnight at 25°C and 170 rpm. The culture was centrifuged and the phage-containing medium was saved. In the second and final third round of panning, 10 µl of MyOne streptavidin T1 beads (65001; Invitrogen) was used to bind 1 µg of biotinylated TACC3, washed, and incubated with 300 µl of phage-containing medium of the previous round for 1 h, then washed five times with a KingFisher robotic platform

(Thermo Fisher Scientific), phage-eluted, and amplified as above.

Phage ELISA was performed, as previously described (Tiede et al., 2014). In brief, 24 randomly picked colonies from the third panning round were used to inoculate 200 µl LB–carbenicillin per well of a 96-well plate, grown, infected with helper phage, and incubated overnight at 25°C and 700 rpm. Plates were centrifuged and 50 µl of phage-containing media was directly applied to TACC3 immobilized wells of 96-well ELISA plate for 1 h at RT. Wells were washed and incubated with anti-Fd-HRP conjugate (A-020-1-HRP; Seramun) for 1 h at RT, washed, and detected with TMB (S-100-TMB; Seramun).

## ELISAs

All proteins were expressed and purified as previously described (Hood et al., 2013; Burgess et al., 2016). 50 µl of 10 µg ml$^{-1}$ biotinylated TACC3 Δ699–765 (diluted in PBS) was immobilized on preblocked HBC Streptavidin plates (Thermo Fisher Scientific) for 30 min at room temperature (RT). Negative control wells were generated by incubating wells with PBS instead of TACC3. Excess TACC3 was removed by washing wells three times with 300 µl PBS and 0.1% TWEEN-20 (PBST). Affimers were diluted in PBS to generate a concentration series of 1 mg ml$^{-1}$ to 0.2 µg ml$^{-1}$ protein. 50 µl of each Affimer sample was applied to the wells and incubated for 1 h at RT. Plates were washed three times with PBST. 50 µl His-HRP antibody (ab1187, 1:5,000; Abcam) was diluted in PBS T20 Superblock (Thermo Fisher Scientific), added to the wells, and incubated for a further 30 min at RT. Plates were then washed three times with PBST. Binding was resolved by the addition of 50 µl TMB (Thermo Fisher Scientific) and quenched by 50 µl 0.5 M sulfuric acid. The well absorbance was read at 450 nm.

## HDX-MS

HDX-MS experiments were conducted using an automated robot (LEAP Technologies) that was coupled to an Acquity M-Class LC with HDX manager (Waters). Samples contained 8.9 mM Na$_2$HPO$_4$, 1.5 mM KH$_2$PO$_4$, 137 mM NaCl, 2.7 mM KCl, pH 7.4. Experiments contained 8 µM TACC3 and 10 µM Affimer. To initiate the HDX reaction, 95 µl of deuterated buffer (8.9 mM Na$_2$HPO$_4$, 1.5 mM KH$_2$PO$_4$, 137 mM NaCl, 2.7 mM KCl, pD 7.4) was transferred to 5 µl of protein-containing solution, and the mixture was subsequently incubated at 4°C for 0.5, 5 or 30 min. Three replicate measurements were performed for each time point and condition studied. 50 µl of quench buffer (8.9 mM Na$_2$HPO$_4$, 1.5 mM KH$_2$PO$_4$, 137 mM NaCl, and 2.7 mM KCl, pH 1.8) was added to 50 µl of the labeling reaction to quench the reaction. The quenched sample (50 µl) was injected onto an immobilized pepsin column (Enzymate BEH; Waters) at 20°C. A VanGuard Pre-column (Acquity UPLC BEH C18 [1.7 µm, 2.1 × 5 mm; Waters]) was used to trap the resultant peptides for 3 min. A C18 column (75 µm × 150 mm; Waters) was used to separate the peptides, employing a gradient elution of 0–40% (vol/vol) acetonitrile (0.1% vol/vol formic acid) in H$_2$O (0.3% vol/vol formic acid) over 7 min at 40 µl min$^{-1}$. The eluate from the column was infused into a Synapt G2Si mass spectrometer (Waters) that was operated in HDMS$^E$ mode. The peptides were separated by ion mobility prior to CID fragmentation in the transfer cell to enable peptide identification. Deuterium uptake was quantified at the peptide level. Data analysis was performed using PLGS (v3.0.2) and DynamX (v3.0.0) (Waters). Search parameters in PLGS were peptide and fragment tolerances = automatic, min fragment ion matches = 1, digest reagent = non-specific, false discovery rate = 4. Restrictions for peptides in DynamX were as follows: minimum intensity = 1,000, minimum products per amino acid = 0.3, maximium sequence length = 25, maximum ppm error = 5, file threshold = 3. Peptides with statistically significant changes in deuterium uptake were identified using the software Deuteros 2.0 (Lau et al., 2021). Deuteros was also used to prepare Woods plots. The raw HDX-MS data have been deposited to the ProteomeXchange Consortium via the PRIDE/partner repository with the dataset identifier, PXD052409 (Perez-Riverol et al., 2022). A summary of the HDX-MS data, as per recommended guidelines (Masson et al., 2019), is shown in Table S1.

## Cell biology

HeLa cells (HPA/ECACC 93021013) or GFP-FKBP-TACC3 knock-in HeLa cells (Ryan et al., 2021) were cultured in DMEM with GlutaMAX (Thermo Fisher Scientific) supplemented with 10% FBS and 100 U ml$^{-1}$ penicillin/streptomycin. Cells were kept at 37°C and 5% CO$_2$.

DNA transfection was by GeneJuice (Merck), according to the manufacturer's instructions; using a 3:1 ratio of transfection reagent to DNA. Typically, cells were processed 48 h after transfection. Aurora A inhibitor MLN8237 (Stratech Scientific) was used at 0.3 µM for 40 min. For induced relocation experiments, rapamycin (Alfa Aesar) was added to cells at a final concentration of 200 nM, 30 min prior to fixation.

## In situ proximity ligation assay

HeLa cells were cultured in the appropriate media and transfected with expression vectors for mCherry or mCherry Affimers for 48 h before fixation. Cells were fixed with ice-cold 100% methanol for at least 30 min at –20°C. Cells were incubated with 3% BSA in PBS (blocking buffer) for 1 h and then with the indicated antibodies diluted in blocking buffer overnight. Mouse anti-TACC3 (1:100, MA525123; Thermo Fisher Scientific) and Rabbit anti-ch-TOG (1:500, PA559150; Thermo Fisher Scientific) were used. Duolink proximity ligation assays were carried out according to the manufacturer's instructions (Duolink; Sigma-Aldrich). The average number of PLA dots detected per cell was calculated using DotCount software, and data represent the mean of four independent experiments ± standard deviation.

## Western blotting

HeLa cells expressing GFP-TACC3 and indicated mCherry-Affimers or mCherry were lysed on ice for 30 min using lysis buffer (10 mM Tris/HCl pH 7.5, 150 mM NaCl, 0.5 mM EDTA, 0.5% Nonidet P40 substitute, 0.09% sodium azide, protease and phosphatase inhibitor tablets). Expression of the target protein(s) was assessed by SDS-PAGE followed by western blotting. The following primary antibodies were used: TACC3 (1:1,000, ab134154; Abcam), ch-TOG (1:1,000, 34032; QED Bioscience), and

mCherry (1:1,000, ab167453; Abcam). This was followed by HRP-conjugated secondary antibodies (1:10,000), and detection was by enhanced chemiluminescence.

## Immunofluorescence

Cells were fixed with PTEMF (20 mM PIPES, pH 6.8, 10 mM EGTA, 1 mM MgCl$_2$, 0.2% Triton X-100, and 4% paraformaldehyde) for 10 min at room temperature or with ice-cold methanol for 10 min. Following permeabilization in 0.5% Triton X-100 in PBS, cells were washed three times with PBS and then blocked at room temperature in 3% BSA in PBS for 1 h and then incubated with primary antibodies diluted in blocking buffer for 1 h. Cover slips were washed three times with PBS three times before incubation with AlexaFluor-conjugated secondary goat antibodies (Invitrogen) in a blocking buffer for 1 h at room temperature. In experiments where CRISPR GFP-FKBP knock-in cell lines were used, anti-Rabbit GFP-boost (Invitrogen) or GFP-boost (Chromotek) antibodies were used to enhance the signal of GFP-tagged proteins. The following antibodies were used: Mouse anti-α-tubulin (1:1,000, T6074; Sigma-Aldrich); Rabbit anti-α-tubulin (1:2,000, PA519489; Invitrogen); Rabbit anti-ch-TOG (1:5,000, 34032; QED Bioscience); Rabbit anti-ch-TOG (1:800, PA5-59150; Thermo Fisher Scientific); Rabbit anti-Pericentrin (1:5,000, ab4448; Abcam); Mouse anti-Centrin-1 (1:500, 04-1624; Sigma-Aldrich); Mouse anti-TACC3 (1:1,000, ab56595; Abcam); anti-Rabbit-GFPboost (1:200, A-21311; Invitrogen); GFPboost (1:200, gba488; Chromotek). Note that during this work, we evaluated the specificity of commercial ch-TOG antibodies for immunofluorescence (Shelford and Royle, 2020).

## Microscopy

Confocal imaging of fixed and live cells was performed using a Nikon CSU-W1 spinning disk inverted microscope equipped with a 2× Photometrics 95B Prime sCMOS camera using either a 100× oil (1.49 NA) or 60× oil (1.40 NA) objective, with respective pixel sizes of 0.110 μm or 0.182 μm. Excitation was sequential, via 405, 488, 561 and 638 nm lasers, with 405/488/561/640 nm dichroic mirrors and Blue, 446/60; Green, 525/50; Red, 600/52; and FRed, 708/75 emission filters. The system also contains an Okolab microscope incubator, Nikon motorized *xy* stage, and a Nikon 200 μm *z*-piezo. Images were acquired with Nikon NI-SElements software.

For mitotic progression experiments, the same microscope system was used, but in widefield mode. A 40× oil (1.30 NA) objective (pixel size, 0.28 μm) was used. Excitation was via a CoolLED (pE-300) light source, with Chroma ZET561/10× (mCherry), Chroma ZET488/10× (GFP), and Chroma ZT647rdc (FRed) excitation filters. Chroma ET575lp (mCherry), Chroma ET500lp (GFP), and Chroma ET665lp and Chroma ZT647rdc (FRed) dichroic mirrors were used with Chroma ET600/50 m (mCherry), Chroma ET525/50 m (GFP), and Chroma ET705/72 m (FRed) emission filters. For live-cell imaging experiments, cells were in 35-mm glass bottom fluorodishes with Leibovitz L15 CO$_2$-independent medium supplemented with 10% FBS at 37°C.

## Image analysis

Analysis was done using Fiji, and the data were exported and read into R for further analysis and plotting. Spindle recruitment analysis was performed as described in Ryan et al. (2021). Using a 1.4-μm² ROI manually placed to measure the average fluorescence intensities of three regions of the spindle (away from the poles), the cytoplasm and one region outside of the cell as background. In R, following background subtraction, the average spindle value was divided by the average cytoplasm value to generate a spindle enrichment ratio and plotted.

To measure spindle morphology and positioning in 3D, a semiautomated procedure was used. First, image stacks were segmented (mCherry channel) using LabKit to obtain a segmented cell volume. The position of centrosomes, a line through the metaphase plate, and markers to delineate the cell of interest were recorded and then, using LimeSeg, a skeleton of the cell was found. In IgorPro, all results were read in and spatial statistics were calculated. A sphere that best fit the surfels generated by LimeSeg was used to calculate the distances from each spindle pole to the cell boundary and to generate the spindle offset measurement. Note that, similar results were obtained in 2D using a routine written in R.

For the quantification of pericentrin and γ-tubulin foci, image stacks were analyzed using 3D Objects Counter in Fiji and subsequent plotting R. Due to the small size of the centrin-1 foci and the tendency for overlap, 3D Objects Counter could not be used, so the counts were done manually by an experimenter blind to the conditions of the experiment. For mitotic progression experiments, the transition stages and the number of γ-tubulin foci were recorded manually and analyzed in R (Sankey diagram) or IgorPro (Mitotic Timing). Transitions were defined as follows: G2–prometaphase, evidence of nuclear envelope breakdown; prometaphase–metaphase, all chromosomes approximately aligned at the metaphase plate; metaphase–anaphase, chromosome segregation.

## Statistical analysis

To compare among three or more groups, a one-way ANOVA was used with Tukey's post hoc test. A Kruskal–Wallis test with Dunn's post hoc test was used for data that did not follow a normal distribution. To assess normality, a Shapiro–Wilk test was used. Fisher's exact test was used to test for an association between Affimer expression and PCM fragmentation. The Bonferroni correction method was used to adjust P values to account for multiple comparisons.

## Online supplemental material

Fig. S1 shows biophysical characterization of TACC3 629-838. Fig. S2 shows NMR studies of ch-TOG 1817–1957. (A) Variability plot of the top 20 ch TOG 1817–1957 structures. Fig. S3 shows NMR interaction and AlphaFold2 modeling of TACC3 and ch TOG, the effect of ch TOG H5 mutations on TACC3 binding and HDX-MS of TACC3 in the absence/presence of Affimers. Fig. S4 shows TACC3 Affimers cannot be used to localize or inducibly relocalize endogenous TACC3 but do affect the mitotic spindle localization of clathrin or TACC3. Fig. S5 shows TACC3 Affimers do not affect mitotic spindle morphology or positioning in HeLa

cells, and PCM fragmentation is not rescued by microtubule depolymerization. Table S1 shows a summary of HDX experimental data.

## Data availability

The NMR co-ordinates of ch-TOG 1817–1957 have been deposited in the RCSB PDB and BMRB databases, with the identifiers PDB: 9F4C; BMRB: 34916. The mass spectrometry proteomics data have been deposited to the ProteomeXchange Consortium via the PRIDE partner repository with the dataset identifier PXD052409. All code used in the manuscript is available at https://github.com/quantixed/p062p035.

## Acknowledgments

EPR experiments were carried out at the EPSRC National Research Facility (NS/A000055/1). NMR experiments were recorded at the KCL Centre for Biomolecular Spectroscopy with the help of RA Atkinson. We thank Claire Mitchell and Laura Cooper from the Computing and Advanced Microscopy Unit (CAMDU) for their help and support.

The work was supported by a Programme Award from Cancer Research UK (C25425/A27718) and Project Awards from BBSRC (BB/L023113/1) and Medical Research Council (MR/X008673/1). J. Shelford was supported by a studentship from the Biotechnology and Biological Sciences Research Council Midlands Integrative Biosciences Training Partnership (MIBTP) (BB/M01116X/1). A.N. Calabrese acknowledges support of a Sir Henry Dale Fellowship jointly funded by the Wellcome Trust and the Royal Society (grant number 220628/Z/20/Z). Funding from the BBSRC (BB/M012573/1) enabled the purchase of HDX-MS equipment.

Author contributions: J. Shelford: Formal analysis, Investigation, Software, Visualization, Writing - original draft, Writing - review & editing, S.G. Burgess: Formal analysis, Investigation, Methodology, Project administration, Resources, Validation, Visualization, Writing - original draft, Writing - review & editing, E. Rostkova: Conceptualization, Data curation, Formal analysis, Investigation, Resources, M.W. Richards: Formal analysis, Investigation, Resources, Visualization, Writing - original draft, Writing - review & editing, G. Larocque: Investigation, J. Sampson: Investigation, C. Tiede: Investigation, Resources, Validation, A.J. Fielding: Formal analysis, Investigation, T. Daviter: Formal analysis, Validation, D.C. Tomlinson: Conceptualization, Methodology, Project administration, Resources, Supervision, Writing - review & editing, A.N. Calabrese: Data curation, Formal analysis, Investigation, M. Pfuhl: Formal analysis, Investigation, Supervision, R. Bayliss: Conceptualization, Funding acquisition, Supervision, Writing - review & editing, S.J. Royle: Funding acquisition, Software, Supervision, Visualization, Writing - original draft, Writing - review & editing.

Disclosures: All authors have completed and submitted the ICMJE Form for Disclosure of Potential Conflicts of Interest. D.C. Tomlinson reported personal fees from Avacta Life Sciences outside the submitted work; in addition, D.C. Tomlinson had a patent to WO2014125290 issued "Avacta Life Sciences." No other disclosures were reported.

Submitted: 1 July 2024

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

# Supplemental material

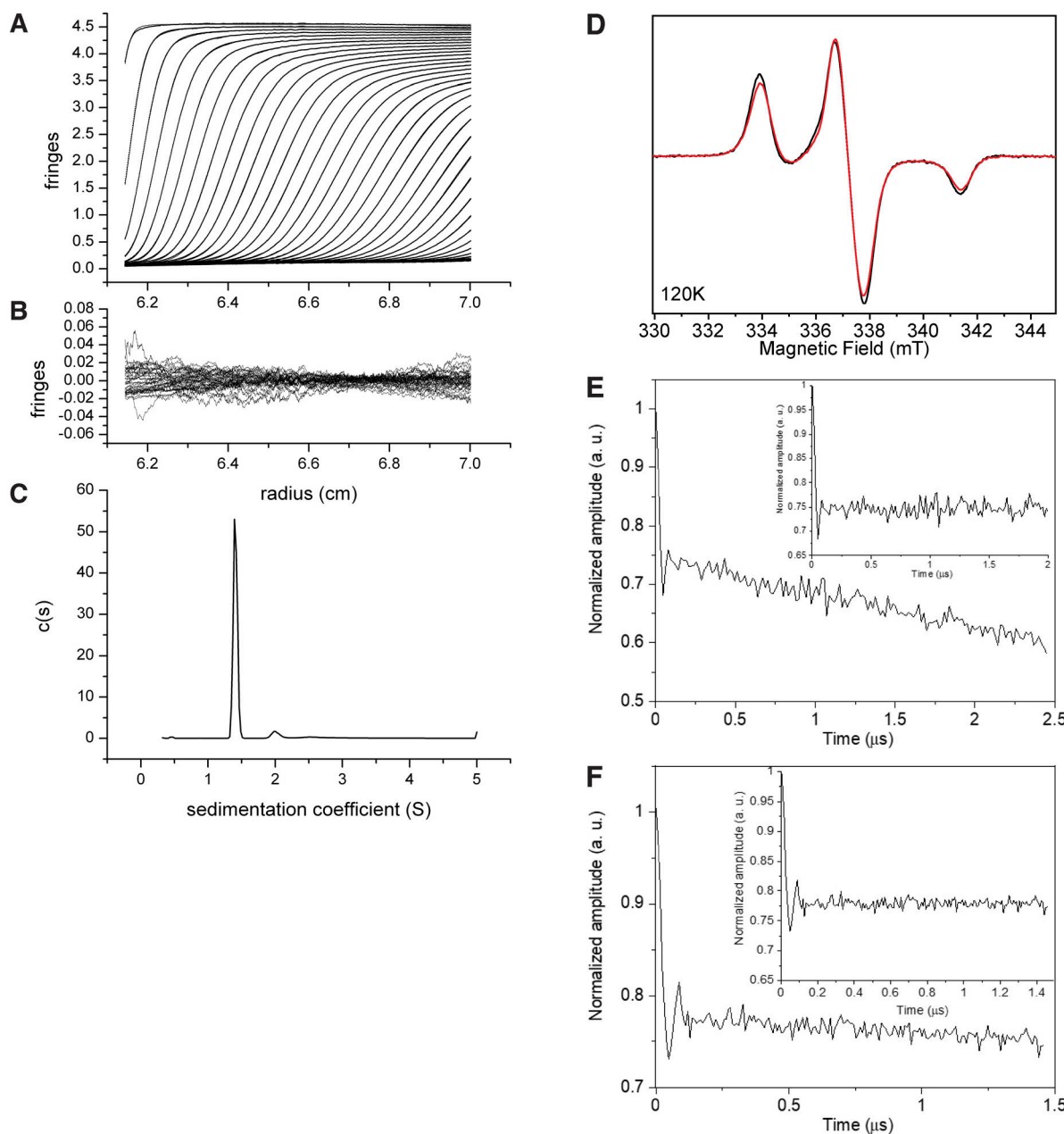

Figure S1. **Biophysical characterization of TACC3 629–838. (A–C)** Sedimentation velocity traces and fits of TACC3 629–838 (2 mg/ml). **(A)** Fitting model: c(s)+1 discrete component, ME regularization at a confidence level of 0.68. **(B)** Residuals of fit shown in A. **(C)** Resulting c(s) distribution. **(D–F)** CW EPR and DEER of TACC3 TACC domain. **(D)** CW EPR spectra of TACC3 MTSL-C828 (black) and TACC3 MTSL-C662 (red) at 120 K. **(E and F)** Normalized four-pulse DEER trace at 50 K for **(E)** TACC3 MTSL-C828 and (F) TACC3 MTSL-C662. Inset, traces after subtraction of a mono-exponential decay. The weak oscillation is most likely from residual proton modulation.

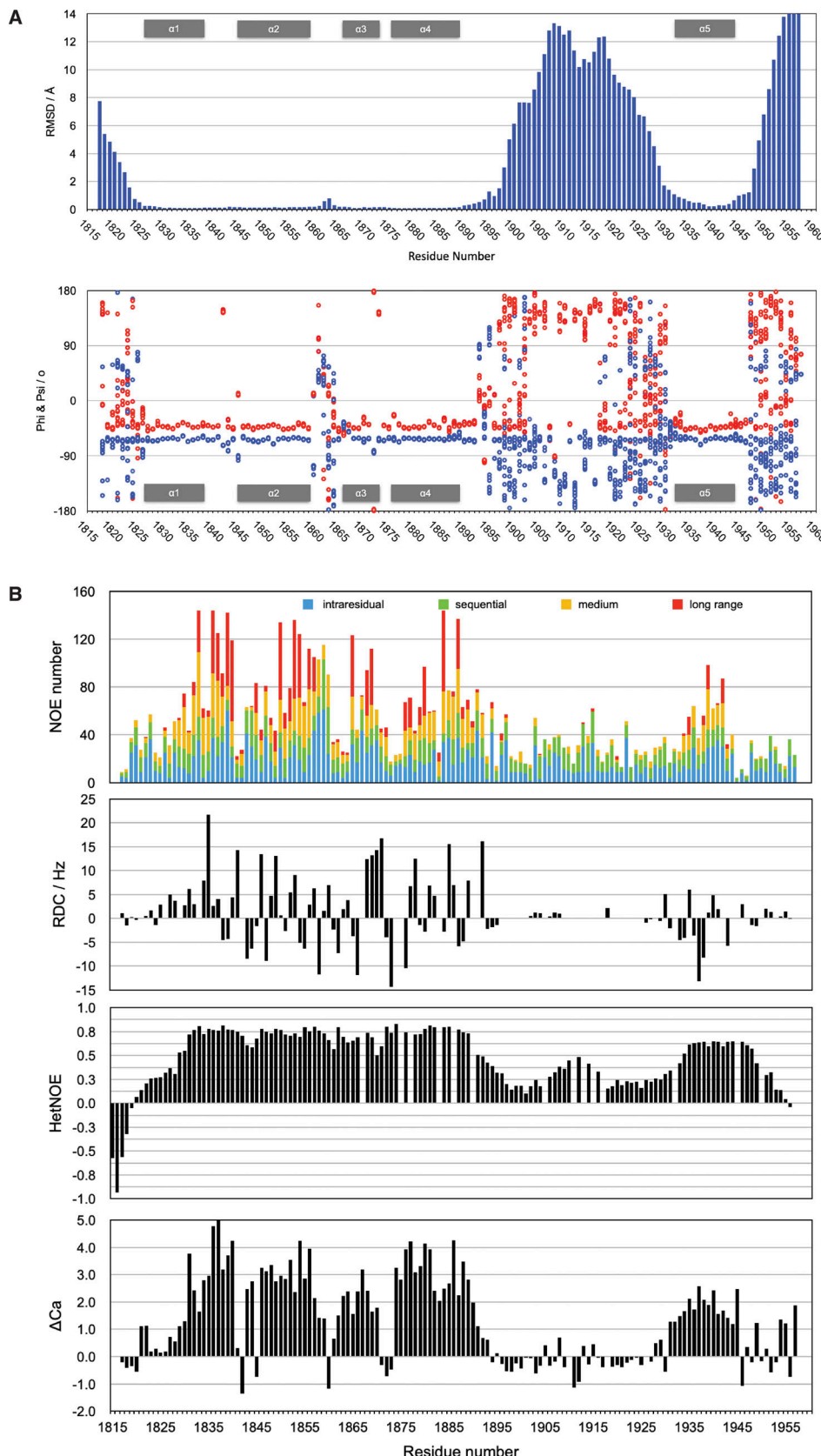

Figure S2. **NMR studies of ch-TOG 1817–1957. (A)** Variability plot of the top 20 ch-TOG 1817–1957 structures. Plots were based on backbone RMSD values (top) and a distribution of φ (blue) and ψ angles (red) (bottom) against amino acid sequence. **(B)** Conformation and stability of ch-TOG 1817–1957. The panels are from top to bottom: number of NOE derived distance constraints, backbone amide RDC values, heteronuclear NOE, and Cα secondary chemical shifts, taken from Rostkova et al. (2018).

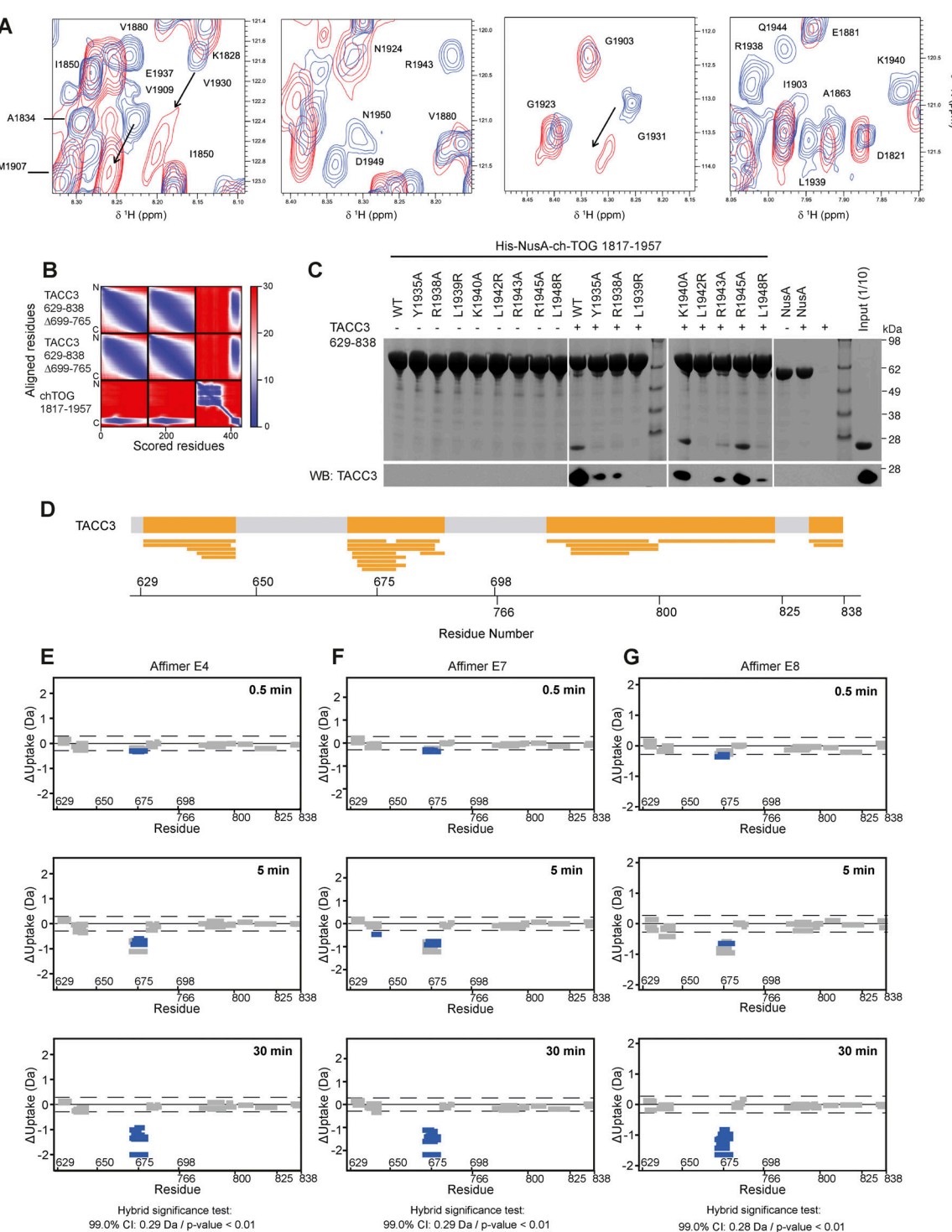

Figure S3.  **NMR interaction and AlphaFold2 modeling of TACC3 and ch-TOG, the effect of ch-TOG H5 mutations on TACC3 binding and HDX-MS of TACC3 in the absence/presence of Affimers. (A)** Portions of a TROSY experiment of $^2$H/$^{15}$N labeled ch-TOG 1817–1957. Plots show the protein alone (blue) and in the presence of a threefold excess of TACC3 629–838 Δ699–765 (red). **(B)** Predicted aligned error (PAE) plot for a model of the complex between TACC3 629–838 Δ699–765 and ch-TOG 1817–1957 generated by AlphaFold2 Multimer. **(C)** In vitro co-precipitation assays between immobilized His-NusA-ch-TOG 1817–1957 constructs and TACC3 629–838 (top). Binding of TACC3 was resolved by western blot (bottom). **(D)** Sequence coverage map of TACC3 629–838 Δ699–765 in HDX-MS experiments. The yellow shaded regions in the thick bar at the top of the panel represent regions with sequence coverage while gray indicates regions that were not covered by detected peptides. Narrow yellow bars represent the individual peptides detected. **(E–G)** Woods plots showing the differences in deuterium uptake in TACC3 at three HDX timepoints (0.5, 5, 30 min of HDX), comparing TACC3 629–838 Δ699–765 alone with TACC3 629–838 Δ699–765 in the presence of Affimers E4 (E), E7 (F) and E8 (G). Woods plots were generated using Deuteros 2.0. Peptides colored in blue are protected from hydrogen/deuterium exchange in the presence of Affimers. Peptides with no significant difference in exchange between conditions, determined using a 99% confidence interval and a hybrid statistical test (dotted line), are shown in gray. A summary of key details of the HDX-MS experiment is shown in Table S1. Source data are available for this figure: SourceData FS3.

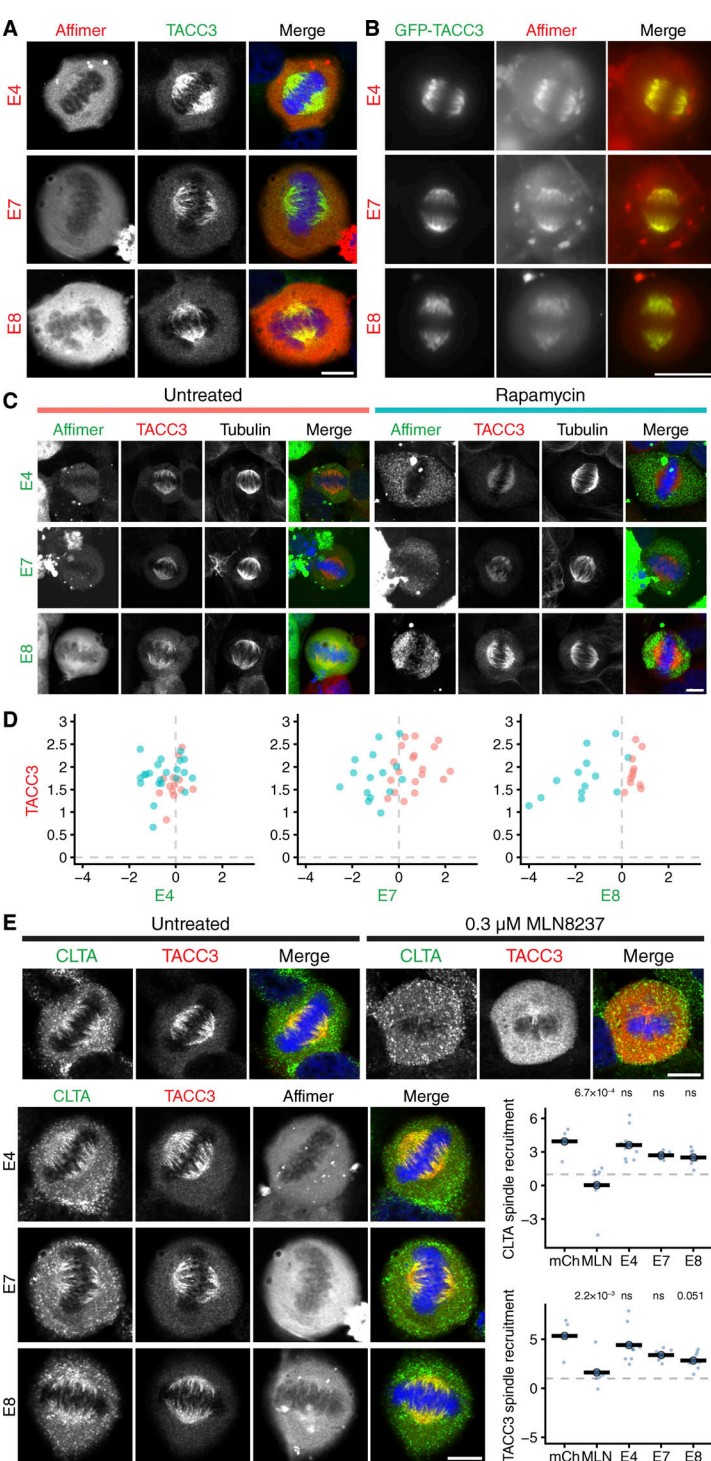

Figure S4. **TACC3 Affimers cannot be used to localize or inducibly relocalize endogenous TACC3 but do affect the mitotic spindle localization of clathrin or TACC3. (A)** Representative confocal micrographs of metaphase HeLa cells expressing the indicated mCherry-Affimers (red). Cells were fixed and stained with anti-TACC3 (green). **(B)** Widefield micrographs of live HeLa cells in metaphase expressing GFP-TACC3 (green) and mCherry-Affimers (red). **(C)** Induced relocalization of TACC3 Affimers to mitochondria. Representative confocal micrographs of HeLa cells at metaphase expressing the indicated FKBP-GFP-Affimers (green) with dark-MitoTrap, that were either treated or not with rapamycin (200 nM, 30 min) prior to fixation. Cells were stained for tubulin (not shown in merge) and TACC3 (red). DNA (blue) is shown in the merge. Relocalization of the Affimer to mitochondria can be seen in the rapamycin-treated cells compared to control, but no relocation of TACC3 is observed, therefore no inactivation of TACC3 activity at the mitotic spindle. **(D)** Quantification of Affimer (x-axis) and TACC3 (y-axis) spindle localization in untreated cells (salmon) and rapamycin treated cells (turquoise). Spindle localization was calculated as the ratio of spindle to cytoplasmic fluorescence shown on a $\log_2$ scale, $n$ = 11–22 cells per condition. **(E)** Representative confocal micrographs of untreated or MLN8237-treated (0.3 µM, 40 min) knock-in CLTA-FKBP-GFP HeLa cells at metaphase. Cells were fixed in PTEMF and stained for TACC3 (red), DNA (blue), and a GFP-boost antibody was used to enhance the signal of CLTA-FKBP-GFP (green). Cells expressing the indicated mCherry-Affimers (gray, not shown in merge), and quantification of spindle recruitment of clathrin (CLTA) and TACC3. Scale bars, 10 µm.

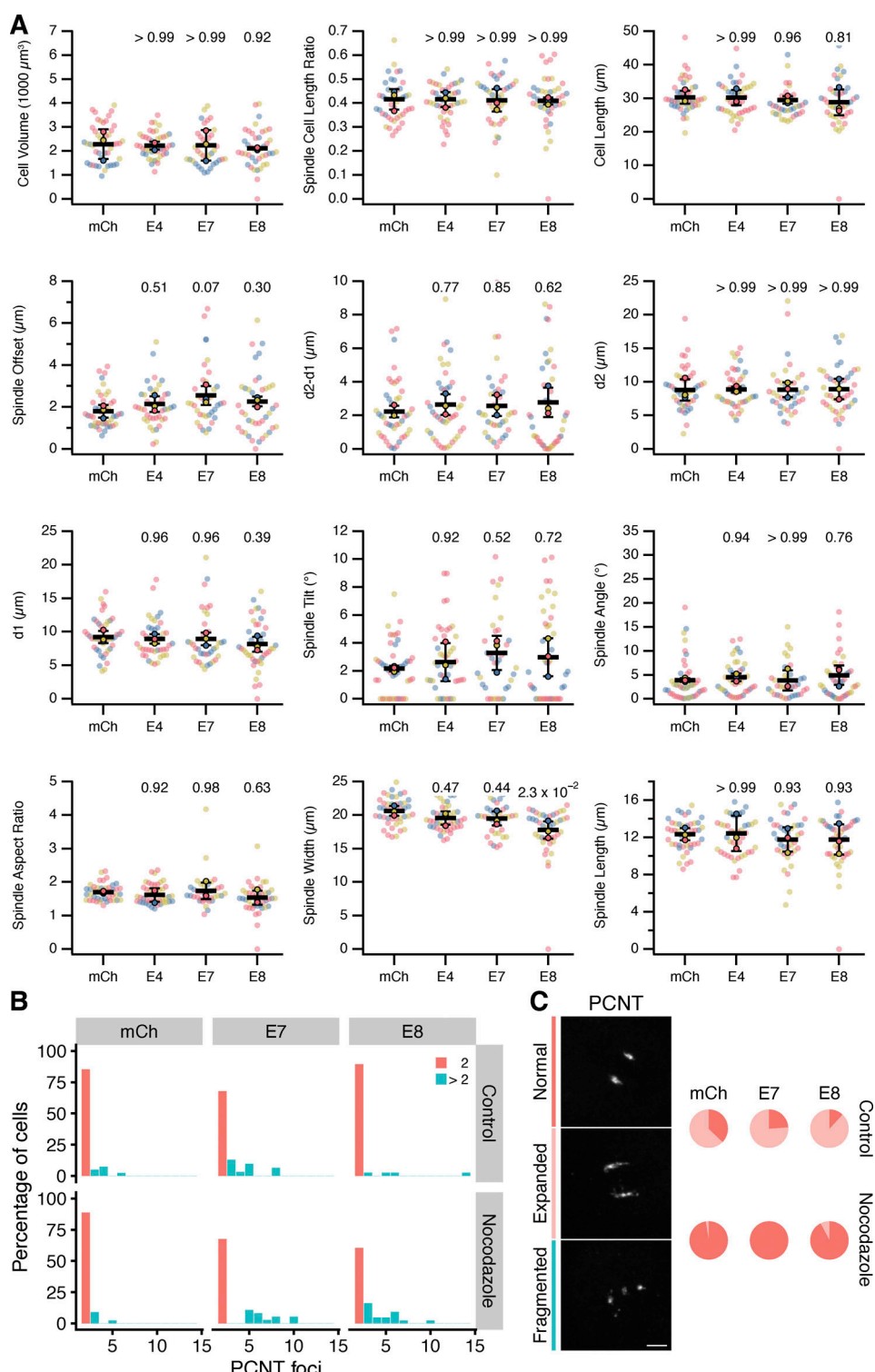

**Figure S5. TACC3 Affimers do not affect mitotic spindle morphology or positioning in HeLa cells, and PCM fragmentation is not rescued by microtubule depolymerization. (A)** Quantification of mitotic spindle parameters in HeLa cells expressing mCherry (mCh) or mCherry-Affimers (E4, E7, E8). Cells were fixed in PTEMF and stained for α-tubulin, pericentrin and DNA. Superplots show the spindle parameters that were measured using a semi-automated workflow. Spindle offset is the euclidean distance between the cell center and the spindle center. The distances d2 and d1 refer to the distance from each centrosome to the cell boundary (taken as a sphere that best fit the 3D perimeter of the cell). Spindle tilt and angle are the angle between the spindle axis and the imaging plane or the metaphase plate, respectively. Dots, single cell; outlined markers, mean independent experiments (indicated by color). Bars show overall mean ± SD; P values from Tukey's HSD post hoc test. **(B)** Counts of PCNT foci in cells expressing mCherry (mCh), or mCherry-Affimers (E7 and E8) for 24 h. Nocodazole treatment (5 µM, 10 min) did not reduce the number of excess PCNT foci. **(C)** Nocodazole was active in these experiments. Example micrographs of PCNT foci show that in two PCNT cells was normal or expanded; while fragmented are cells with >2 PCNT foci. Scale bar, 10 µm. Pie charts of the fraction of 2 PCNT cells that had normal or expanded PCNT staining.

**Provided online is Table S1. Table S1 shows HDX data summary table.**

