## [Peer Review File · The Journal of Cell Biology]

Structural characterization and inhibition of the interaction between ch-TOG and TACC3

James Shelford, Selena Burgess, Elena Rostkova, Mark Richards, Gabrielle Larocque, Josephina Sampson, Christian Tiede, Alistair Fielding, Tina Daviter, Darren Tomlinson, Antonio Calabrese, Mark Pfuhl, Richard Bayliss, and Stephen Royle

Corresponding Author(s): Stephen Royle, University of Warwick and Mark Pfuhl, King's College London

Review Timeline:

Submission Date:	2024-07-01
Editorial Decision:	2024-08-20
Revision Received:	2024-12-19
Editorial Decision:	2025-02-03
Revision Received:	2025-02-12

Monitoring Editor: Monica Bettencourt-Dias

Scientific Editor: Tim Fessenden

Transaction Report:

DOI: <https://doi.org/10.1083/jcb.202407002>

August 20, 2024

Re: JCB manuscript #202407002

Prof. Stephen J Royle
University of Warwick
Division of Biomedical Sciences
Centre for Mechanochemical Cell Biology
Gibbet Hill Road
Coventry CV4 7AL
United Kingdom

Dear Prof. Royle,

Thank you for submitting your manuscript entitled "Structural characterization and inhibition of the interaction between ch-TOG and TACC3". The manuscript was assessed by expert reviewers, whose comments are appended to this letter. Thank you for your patience while we sought reviewer feedback and considered a decision, which was delayed due to editor availability. We invite you to submit a revision if you can address the reviewers' key concerns, as outlined here.

As you will see, reviewers commended the structural insight into TACC3 and ch-TOG interactions provided by this work. All reviewers remarked, in some form, that the conceptual link between the structural data and the observations in cells should be clarified and extended. A suitably revised manuscript should focus on this connection as laid out particularly by Reviewers 1 (points 3-5) and Reviewer 2 (points 2 and 6). In their second point, Reviewer 3 offers further suggestions for improving the observations in cells. While all reviewer comments should be addressed in some form, additional data beyond these points is not required in a revision.

GENERAL GUIDELINES:

Text limits: Character count for an Article is < 40,000, not including spaces. Count includes title page, abstract, introduction, results, discussion, and acknowledgments. Count does not include materials and methods, figure legends, references, tables, or supplemental legends.

Figures: Articles may have up to 10 main text figures. Figures must be prepared according to the policies outlined in our Instructions to Authors, under Data Presentation, <https://jcb.rupress.org/site/misc/ifora.xhtml>. All figures in accepted manuscripts will be screened prior to publication.

*****IMPORTANT:** It is JCB policy that if requested, original data images must be made available. Failure to provide original images upon request will result in unavoidable delays in publication. Please ensure that you have access to all original microscopy and blot data images before submitting your revision. ***

Supplemental information: There are strict limits on the allowable amount of supplemental data. Articles may have up to 5 supplemental figures. Up to 10 supplemental videos or flash animations are allowed. A summary of all supplemental material should appear at the end of the Materials and methods section.

Please note that JCB now requires authors to submit Source Data used to generate figures containing gels and Western blots with all revised manuscripts. This Source Data consists of fully uncropped and unprocessed images for each gel/blot displayed in the main and supplemental figures. Since your paper includes cropped gel and/or blot images, please be sure to provide one Source Data file for each figure that contains gels and/or blots along with your revised manuscript files. File names for Source Data figures should be alphanumeric without any spaces or special characters (i.e., SourceDataF#, where F# refers to the associated main figure number or SourceDataFS# for those associated with Supplementary figures). The lanes of the gels/blots should be labeled as they are in the associated figure, the place where cropping was applied should be marked (with a box), and molecular weight/size standards should be labeled wherever possible.

The typical timeframe for revisions is three to four months. While most universities and institutes have reopened labs and allowed researchers to begin working at nearly pre-pandemic levels, we at JCB realize that the lingering effects of the COVID-19 pandemic may still be impacting some aspects of your work, including the acquisition of equipment and reagents. Therefore, if you anticipate any difficulties in meeting this aforementioned revision time limit, please contact us and we can work with you to find an appropriate time frame for resubmission. Please note that papers are generally considered through only one revision cycle, so any revised manuscript will likely be either accepted or rejected.

Thank you for this interesting contribution to Journal of Cell Biology. You can contact us at the journal office with any questions at cellbio@rockefeller.edu.

Sincerely,

Monica Bettencourt-Dias
Monitoring Editor
Journal of Cell Biology

Tim Fessenden
Scientific Editor
Journal of Cell Biology

Reviewer #1 (Comments to the Authors (Required)):

The paper by Shelford et al. describes the structural details of the interaction of ch-TOG with TACC3, two factors required for mitotic spindle assembly as part of multiple different protein complexes. This insight allowed the authors to develop a molecular tool (Affimer) that is able to specifically block this interaction, allowing them to dissect the specific mitotic function of the ch-TOG-TACC3 complex. Unexpectedly, they found a role for this complex in ensuring pericentriolar material integrity during mitosis.

Overall, the manuscript is well written and the data is of high quality. The structural data is very comprehensive and convincing, and the Affimers derived from this allowed identification of a function that would be otherwise difficult to specifically target. However, I have some concerns regarding the functional analysis of the ch-TOG-TACC3 interaction and the PCM fragmentation phenotype. The mechanistic basis of this phenotype is unclear and has not been investigated. It would be great if the authors could add some work to address this and derive some explanation for this defect.

Major points:

- 1) For protein quantifications in the mitotic spindle (i.e. Figure 5BC), a marker for microtubules should be used to confirm that the reduction of the protein in the spindle is not simply due to a reduction in spindle microtubule density. Normalization of the protein intensity to tubulin intensity would test for this.
- 2) S8: There should be a quantification of the intensity of TACC3 and Clathrin in the spindle.
- 3) The authors should test Affimer expression in a non-cancer cell line such as RPE-1, to test if pole fragmentation is a general effect and not only observed in cells that may have an altered PCM.
- 4) Spindle pole fragmentation: this is a known phenotype for cells undergoing prolonged mitosis and could thus be indirect. For example, this is also observed for augmin depletion, which has no role in PCM assembly or activity (Lawo et al., 2009, Curr Biol). The authors should test whether fragmentation is MT-dependent (can it be rescued by nocodazole treatment?). If so, pole fragmentation may be due to imbalanced spindle forces, rather than a direct role of TACC3-ch-TOG in PCM integrity. This should then also be discussed.
- 5) Related to point 4: the authors showed in their previous work that TACC3 is not at the mitotic PCM. In the current work there

seems to be spindle pole signal in some cases (Fig 5). Considering the PCM fragmentation phenotype, it would be useful to test TACC3 (and perhaps also a tagged version) carefully for localization at mitotic centrosomes, with and without microtubules. This information together with MT-dependency would greatly help with the interpretation of the phenotype.

6) In quantifications of cell measurements (Figure 5C, 6C, 7C, S7C and S9), the authors have pooled single cell measurements of three independent experiments and consider them independent replicates. However, this leads to a very high n number that produces artificially low estimates of variance and thus increases the possibility of a type I error in statistical tests. Additionally, the variable n (i. e. Figure 5C: n=40-45) would also cause an unbalanced weight of each true experimental replicate. It would be better to quantify at a certain number of cells per condition for each of the (3 or more) independent experiments and use the sample means from each of the experiments for statistics.

Minor points:

-The source and purification of the recombinant proteins used should be at least briefly explained in the results

-3B: KD mentioned in text cannot be seen in the graph; also correct axes scale and labels

-the authors should consider switching IF images to color blind-friendly combinations, such as green-magenta-blue or magenta-yellow-cyan.

-There should be an explanation on why the 699-768 deletion for TACC3 is used for the structural model of the interaction.

-How is it that, in Figure 5B, in the control condition where cells express mCherry alone, there is mCherry signal in the spindle?

-Supplementary Figure S7:

a) The letter C for Figure S7C is misplaced and there is no D, while it is described in the caption.

b) If I understood correctly, in S7A, WT HeLa cells were fixed in PTEMF and then the CRISPR GFP-FKBP-TACC3 HeLa cells were fixed in methanol. Is that change only due to the different antibodies used? This should be explained.

-All IF figures: to be consistent, for each channel I would indicate the antibody used on one side and the expressed construct on the other side of the figure. Currently this is inconsistent and confusing in several cases, e.g. Fig. 7A (GFP labeled as chTOG-GFP). Fig. S7A,B,C (various inconsistencies)

-About the paragraph in the Results section that talks about Supplementary Figure S7: "None of the Affimers performed well as reagents for visualization or inducible relocalization of TACC3 (Supplementary Figure S7). However, we were able to see colocalization of each of the three Affimers with overexpressed GFP-TACC3 (Supplementary Figure S7)." I believe these experiments need to be explained in a bit more detail (such as the role of rapamycin) and the significance of the outcome.

Reviewer #2 (Comments to the Authors (Required)):

Structural characterization and inhibition of the interaction between cg-TOG and TACC2

In this manuscript, Shelford et al. investigate the interaction between the TOG domain protein ch-TOG and TACC3. These proteins are components of several common complexes, such as TACC3-clathrin-ch-TOG-GTSE1, which is involved in microtubule cross-linking, and TACC3-ch-TOG, which tracks microtubule plus ends. They also have independent functions; for example, ch-TOG binds to kinetochores via HEC1. The authors highlight the complexity of understanding the specific roles of TACC3 and ch-TOG within different complexes.

To elucidate the structure of the TACC3-ch-TOG complex, the authors conducted extensive biophysical and structural analyses of both proteins individually and in combination. Using a broad range of experimental techniques, including NMR, AUC, EPR, MST, and HDX-MS, they obtained detailed structural insights. These include the identification of the parallel coiled-coil dimer of TACC3 (residues 629-838) and the helical core of the ch-TOG region (residues 1817-1957), as well as the formation of the TACC3-ch-TOG complex.

The Affimer strategy introduced in this study specifically targets TACC3 TD and selectively disrupts the interaction between TACC3 and ch-TOG both in vitro and in vivo. The researchers identified six different Affimer groups, further characterizing E4, E5, E7, and E8. E4, E7, and E8 were found to bind to amino acids 670-682 of TACC3, which corresponds to the ch-TOG binding site. Among these, E8 exhibited the strongest binding to TACC3 and inhibited the formation of the TACC3-ch-TOG complex.

When applied to cells, the Affimer disrupted the interaction between TACC3 and ch-TOG, leading to the mislocalization of ch-TOG from the mitotic spindle without affecting clathrin or TACC3. This allowed the authors to investigate the function of ch-TOG in the mitotic spindle without disrupting TACC3 localization or impairing general ch-TOG functions, which would occur when using a ch-TOG degron. Interestingly, Affimer expression also affected the localization of pericentrin, resulting in additional pericentrin foci in mitotic cells, indicating PCM fragmentation during mitosis without affecting the centrioles (as evidenced by the lack of centrin 1 in the additional pericentrin dots).

This manuscript presents complex data from a wide range of experiments in a competent and well-discussed manner. The Affimer approach is very interesting and the authors provide a very good case for the use of this strategy. The authors demonstrate that the TACC3-ch-TOG complex in the mitotic spindle is disrupted by Affimer expression. However, the transition to PCM fragmentation is abrupt and lacks a clear connection. This section needs improvement to enhance the flow and coherence. The pericentrin fragmentation phenotype is interesting; however, the underlying molecular mechanisms are not addressed. The authors should provide additional data in this direction. Additionally, I have several concerns outlined below that should be addressed.

Major concerns

1. It is impressive that authors applied a broad range of different techniques and spent quite some effort to describe the structure characterization of both TACC3 TD and ch-TOG 1817-1957 region in detail. This may be attractive to a special group of people in the field but could also have the potential risk to lose the interest of the board readers, especially given that these data are consistent with the previous models (based AlphaFold2 and deletion analysis) and also not considered as the major focus of this study. Therefore, it makes sense if the authors shorten this result section and combine the first three sections.
2. The fragmentation of centrosome PCM caused by the disruption of TACC3 and ch-TOG interaction via Affimers treatment is a relatively novel phenotype and one of the key findings in this study. However, the direct connection between the TACC3 and ch-TOG interaction and the PCM integrity during mitosis is missing. Is the TACC3-ch-TOG complex also detected in the PCM or centrosome region during mitosis? Would the recruitment and/or activation of γ -TuRC in PCM also impaired upon the Affimers treatment? More related discussion or interpretation will also help to better understand this interaction.
3. Across the study, the truncated version of TACC3, TACC3 629-838 Δ 699-765 (TACC3 TD Δ), is used for the complex structure prediction, Affimer generation, and several in vitro binding assays. It is worth having a more detailed introduction of this truncated version and why it was chosen in this study.
4. The Affimer is crucial in this study to target TACC3 and disrupt the interaction between TACC3 and ch-TOG. Authors did extensive characterization of these Affimers both in vitro and in vivo, but it also raised some concerns, like the Affimer could not co-localized well with endogenous TACC3 very well in vivo (but works better with the overexpressed TACC3), the selectivity of Affimers in cell (which could be further addressed by doing the cell lysate pulldown using resin-immobilized Affimer), and decrease of TACC3 spindle recruitment upon Affimers treatment (Figure 5C-D, which may also indicate some other functions of these Affimers in cells). Authors should also address (could be non-experimentally) the pros and cons of Affimer approach compared to some other wilder used methods, like partial deletion of the binding site or introducing the binding deficient mutants for either protein.
5. In Figure 4A, Affimer is used to be immobilized on the resin and pulldown other proteins, but its signal in the first E7 co-precipitation sample is much lower compared to others. Meanwhile, some extra bands (I would assume they are minus impurities from protein purification) could be observed in the TACC3 TD sample. Some special labels of these bands and/or discussion in the text would be appreciated. Later in the same experiment, TACC3 TD is used to pulldown ch-TOG as a positive control, but based on the previous experimental settings (a. The Affimers were generated based on TACC3 TD Δ . b. in the early part of the same experiment, only TACC3 TD Δ was used together with ch-TOG), TACC3 TD Δ should also be included and tested its interaction with ch-TOG.
6. From the current data, the phenotype of Affimer treatment is similar to the depletion of ch-TOG, which indicates that the ch-TOG-TACC3 complex is important for PCM integrity. Authors could check if Affimer treatment affects the ch-TOG level in general (using immunoblot) and around in PCM specifically (using immunofluorescence).

Minor comments

1. Fig. 4D does not show inhibition of ch-TOG binding to TACC3 in vitro as stated in the text.
2. Since the author put the gene name of ch-TOG, CKAP5, in the keyword, it is better to have a brief introduction of it in the text for the general audience.
3. The construct information of NusA-TACC3 TACC domain is missing in the methods section.
4. In Figure 1A, the second label on the left (ch-TOG 1467-2032/NusA) is supposed to indicate two different bands on two different gels, but might be a little bit misleading, it might be better to separate the label for each gel respectively.
5. A consistent color-coding for the five helices of ch-TOG 1817-1957 in all of the figures (especially in figures 1, 2, and 3) could be helpful to deliver the information to the readers.
6. In Figure 3A, the label of the y-axis could be changed to CSPs (ppm), which might be more friendly to the general audience. The 1* and 2* labels in the spectrum also require further description in the legend.
7. The axis labels are missing for Figure S5.
8. Did the author also try to predict the structure of ch-TOG 1817-1957 region alone using AlphaFold? Does H5 more prefer to dock on the H1-H4 core or rather flee out freely?
9. In Figure S7, based on the legend, the panel D is missing.

10. In Figure 5B, the representative images of mCherry control don't look very consistent with the quantification results (Fig 5C), so it would be better to update with another image.
11. In Figure S8, further quantification is required to demonstrate clathrin is not impacted by the Affimers treatment.

Reviewer #3 (Comments to the Authors (Required)):

In this manuscript, Shelford and colleagues focus on two important regulators of spindle assembly, TACC3 and ch-TOG. First, they perform a structural characterization of the complex, providing structural and molecular detail for the TACC3/ch-TOG interaction. Building on these bases, they identify Affimers with the ability to bind TACC3 and interfere with ch-TOG interaction. They go on characterizing the cellular effects of identified Affimers, in order to dissect the mitotic functions that are directly dependent on the TACC3/ch-TOG complex. The work addresses an important point, since dissecting the roles of specific subpools of multi-functional proteins such as TACC3 and ch-TOG is an interesting and challenging objective. Authors use approaches going from computational, biochemical and biophysical to cellular assays, experimental data are solid and overall clearly presented. Still, I recommend additional experiments to be performed prior to publication, to strengthen the conclusions and widen the obtained information. Specific points are listed below.

1. The manuscript describes a structural model of the interface between TACC3 and ch-TOG, generated using AlphaFold2-Multimer. However, this approach has several limitations that should be addressed, in particular: a) While AlphaFold2 was a groundbreaking tool in protein structure prediction, AlphaFold3 has since been released and has demonstrated superior performance in predicting protein structures, especially in multimeric contexts. The use of AlphaFold2 instead of AlphaFold3 might result in less accurate or less reliable models. The authors should consider updating their methodology to incorporate the latest version to enhance the accuracy and confidence in their predictions. b) The predictions are based on specific fragments of TACC3 (residues 629-838 Δ 699-765) and ch-TOG (residues 1817-1957). While focusing on fragments can simplify the modeling process and reduce computational load, it may overlook important interactions and conformational changes that occur in the context of the full-length proteins. Full-length protein analysis, at least at the computational level, would provide a more comprehensive view of the interface and its functional implications.
2. The Affimer binding to TACC3 and the impaired interaction between TACC3 and ch-TOG are key results on which the interpretation of cellular assays is based. As such, I find the experiment in Fig. 4E is not sufficiently informative. Furthermore, I find confusing that Affimers E7 and E8 display (Fig. 4E) such different binding ability to TACC3 and different capability to interfere with ch-TOG binding to TACC3, but highly similar phenotypes in the following assays. a. The disruption of the TACC3/chTOG complex in single cells at specific mitotic stages should be demonstrated by isPLA, or other proximity-based, assays and endogenous components. This would provide a measure of how many mitotic cells, and to which extent, display reduced interaction. Parallel PLA assays using mCherry and TACC3 (or GFP) antibodies would enable authors to evaluate and compare to which extent and where the Affimers/target interaction occurs, since localization patterns are not too clear (S7). b. Dose response assays (with different amounts of transfected Affimers would also be useful for a better understanding of Affimers mode of action.
3. Although the TACC domain is expected not to be directly involved in the mechanisms related to the AurkA/TACC3 interplay (AurkA activation, TACC3 phosphorylation), cellular scenarios may be more complex, and I feel it would be important to show that AurkA activation (and potentially TACC3 phosphorylation) is not modified in cells upon Affimers transfection. Related to this, it is not clear to me how authors explain the decrease of TACC3 at the spindle.
4. It appears that clathrin localization is an important result to show the specificity of action of the Affimers towards the TACC3/ch-TOG interaction and function. Images in Fig. S8 should therefore be accompanied by a quantification analysis.
5. Some additional quantifications under the experimental conditions of Figures 5-6 would help interpreting data and add relevant information: a. how is the cell-to-cell variability of the mCherry Affimer signal? b. which is the amount of spindle tubulin upon treatment? c. Is ch-TOG localization altered in prometaphase too?
6. Authors propose force unbalances to underlie to observed fragmentation. Rescue experiments by altering spindle forces in independent manners would support this hypothesis and would add information about the mechanism through which this chTOG/TACC3 function is exerted.
7. Are Affimers able to bind TACC1 and TACC2, and can this contribute to the phenotypes?

Minor

1. Fig. 6E. If I correctly understand the display mode of the plot, there are 5 cells with 2 centrioles in each spot, out of the 21 with supernumerary PCM spots (Affimer 8). Would this imply that both PCM fragmentation and centrosome amplification occur, although to different extents?

2. Could authors explain in more detail how the time lapse analysis is performed, in particular how "transitions" are defined?
3. A recent paper (Rajeev et al., 2023, mentioned in the Introduction) has addressed this question by a different approach, similar to the one used in Fig 7. Authors should comparatively comment the results observed by the different approaches.

Reviewer #1 (Comments to the Authors (Required)):

The paper by Shelford et al. describes the structural details of the interaction of ch-TOG with TACC3, two factors required for mitotic spindle assembly as part of multiple different protein complexes. This insight allowed the authors to develop a molecular tool (Affimer) that is able to specifically block this interaction, allowing them to dissect the specific mitotic function of the ch-TOG-TACC3 complex. Unexpectedly, they found a role for this complex in ensuring pericentriolar material integrity during mitosis.

Overall, the manuscript is well written and the data is of high quality. The structural data is very comprehensive and convincing, and the Affimers derived from this allowed identification of a function that would be otherwise difficult to specifically target. However, I have some concerns regarding the functional analysis of the ch-TOG-TACC3 interaction and the PCM fragmentation phenotype. The mechanistic basis of this phenotype is unclear and has not been investigated. It would be great if the authors could add some work to address this and derive some explanation for this defect.

We thank the Reviewer for their time and comments on our manuscript. We were very pleased with their positive assessment of our work.

Major points:

1) For protein quantifications in the mitotic spindle (i.e. Figure 5BC), a marker for microtubules should be used to confirm that the reduction of the protein in the spindle is not simply due to a reduction in spindle microtubule density. Normalization of the protein intensity to tubulin intensity would test for this.

*The data presented show the spindle recruitment (spindle / cytoplasm), but the concern is that a reduction in spindle microtubule density could artificially reduce the spindle recruitment values. We have now tested this by measuring tubulin intensity at metaphase spindles in cells expressing mCherry or mCherry-tagged Affimers E4, E7 or E8. There is no measurable change between conditions, $p = 0.818$. This data has been added to **Figure 5**.*

2) S8: There should be a quantification of the intensity of TACC3 and Clathrin in the spindle.

*Quantification of the spindle recruitment of TACC3 and clathrin has been added to this Figure, which is now **Figure S4**.*

3) The authors should test Affimer expression in a non-cancer cell line such as RPE-1, to test if pole fragmentation is a general effect and not only observed in cells that may have an altered PCM.

*We had issues transfecting RPE-1 at high enough efficiency in order to analyze sufficient numbers of mitotic cells. To address this point, we expressed the Affimers in HEK293T cells and found no evidence of fragmentation. This suggests that the phenotype is specific to cells with altered PCM. This result has been added to the **Results** section text. "This phenotype may be specific to cells with altered PCM because HEK293T cells did not show the fragmentation (2 foci in 80% of cells, across conditions)."*

4) Spindle pole fragmentation: this is a known phenotype for cells undergoing prolonged mitosis and could thus be indirect. For example, this is also observed for augmin depletion, which has no role in PCM assembly or activity (Lawo et al., 2009, Curr Biol). The authors should test whether fragmentation is MT-dependent (can it be rescued by nocodazole treatment?). If so, pole fragmentation may be due to imbalanced spindle forces, rather than a direct role of TACC3-ch-TOG in PCM integrity. This should then also be discussed.

We have tested this point by examining whether the PCNT foci frequency in cells expressing Affimers is reduced by nocodazole. We found no evidence for rescue. We used tubulin staining to confirm that this treatment was sufficient to depolymerize microtubules.

Additionally, we could see a change in the centriolar PCM arrangement in nocodazole-treated cells, so the treatment was apparently sufficient to decrease forces on the poles. These results are now in **Figure S5B** and are described in the text. We have removed the sentence in the Discussion referring to the fragmentation being caused by microtubule forces.

5) Related to point 4: the authors showed in their previous work that TACC3 is not at the mitotic PCM. In the current work there seems to be spindle pole signal in some cases (Fig 5). Considering the PCM fragmentation phenotype, it would be useful to test TACC3 (and perhaps also a tagged version) carefully for localization at mitotic centrosomes, with and without microtubules. This information together with MT-dependency would greatly help with the interpretation of the phenotype.

To address this point, we repeated our previous work. The results were reassuringly similar to those reported in Gutiérrez-Caballero et al. 2015. Briefly, TACC3 is adjacent to the mitotic PCM and mainly on spindle microtubules. When microtubules are depolymerized with nocodazole, TACC3 forms a ring with pericentrin at its core. This ring doesn't represent microtubule stubs because it is largely devoid of tubulin (Reviewer Figure 1, below). These observations are in keeping with the idea that TACC3 forms a scaffold adjacent to the PCM. Because these results are already published, we have not included them in the paper. Instead, we now show averaged images of the TACC3/ch-TOG localization in **Figure 5** to highlight that ch-TOG and TACC3 have distinct localizations in mitotic cells.

Reviewer Figure 1

6) In quantifications of cell measurements (Figure 5C, 6C, 7C, S7C and S9), the authors have pooled single cell measurements of three independent experiments and consider them independent replicates. However, this leads to a very high n number that produces artificially low estimates of variance and thus increases the possibility of a type I error in statistical tests. Additionally, the variable n (i. e. Figure 5C: n=40-45) would also cause an unbalanced weight of each true experimental replicate.

It would be better to quantify at a certain number of cells per condition for each of the (3 or more) independent experiments and use the sample means from each of the experiments for statistics.

We thank the Reviewer for this point. We were slightly confused by the comment because an inflated n number would increase the Type II, not Type I, error rate. This not an issue for

most of the figures, which show *p* values greater than 0.05. For these figures, an elevated Type I rate would be a concern, but cannot be due to pooling measurements. The Reviewer's concern that we did not show experimental replicability for these figures is correct. So, we have changed all of them to show SuperPlots which highlight the experimental replicate averages as well as all the data points. The statistical tests are performed now with *n* = 3. The *p*-values are changed but the key findings remain the same. With the exception of the TACC3 reduction which is now not consistently significantly different across conditions (see above).

Minor points:

-The source and purification of the recombinant proteins used should be at least briefly explained in the results

Brief descriptions of the source and purification of TACC3, ch-TOG and Affimers have been added to the relevant Results sections.

-3B: KD mentioned in text cannot be seen in the graph; also correct axes scale and labels

The data has been plotted and fitted by nonlinear regression to a One site - specific binding model in Prism 10 (Graphpad). The fitted K_D is $17.3 \pm 2.0 \mu\text{M}$, and this approximate value can be seen on the graph.

-the authors should consider switching IF images to color blind-friendly combinations, such as green-magenta-blue or magenta-yellow-cyan.

Our solution to accessibility is to show the individual channels as grayscale images that can be seen by all. Red-green-blue is the only option for a three way merge that allows the assessment of overlap between each of the three channels.

-There should be an explanation on why the 699-768 deletion for TACC3 is used for the structural model of the interaction.

The rationale behind the use of TACC3 629-838 Δ 699-765 has been inserted into the manuscript within 'Interaction of ch-TOG 1817-1957 with TACC3'.

-How is it that, in Figure 5B, in the control condition where cells express mCherry alone, there is mCherry signal in the spindle?

The spindle can be infiltrated by mCherry which is only 27 kDa. We often see a slight enrichment of GFP, mCherry or other fluorescent proteins in the spindle region.

-Supplementary Figure S7:

a) The letter C for Figure S7C is misplaced and there is no D, while it is described in the caption.

b) If I understood correctly, in S7A, WT HeLa cells were fixed in PTEMF and then the CRISPR GFP-FKBP-TACC3 HeLa cells were fixed in methanol. Is that change only due to the different antibodies used? This should be explained.

Thank you for spotting this. We have rearranged this figure (now Fig S4) because JCB has a Supplemental Figure limit, and have merged and pruned some panels. To answer the question: it was just two ways of looking at endogenous TACC3 and using different fixatives to show that the result was not dependent on the fixative.

-All IF figures: to be consistent, for each channel I would indicate the antibody used on one side and the expressed construct on the other side of the figure. Currently this is inconsistent and confusing in several cases, e.g. Fig. 7A (GFP labeled as chTOG-GFP). Fig. S7A,B,C (various inconsistencies)

Again, thanks for spotting this. Corrected.

-About the paragraph in the Results section that talks about Supplementary Figure S7: "None of the Affimers performed well as reagents for visualization or inducible relocation of TACC3 (Supplementary Figure S7). However, we were able to see colocalization of each of the three Affimers with overexpressed GFP-TACC3 (Supplementary Figure S7)." I believe these experiments need to be explained in a bit more detail (such as the role of rapamycin) and the significance of the outcome.

The Reviewer is correct that the description of the results was too terse. Essentially Affimers could be used like nanobodies such that we could use them to i) visualise the endogenous TACC3 or ii) mislocalize TACC3 by inducibly relocating the Affimer. Neither of these things was possible, although when GFP-TACC3 was over-expressed, we could use the Affimers to label it. We have changed the sentence to read "Although Affimers have the potential to be used akin to nanobodies, none of the Affimers performed well as reagents for visualization or inducible relocation of endogenous TACC3 (Figure S4A-C)." The section has been reworked because of other changes and now the colocalization result precedes this sentence, which should make it all clearer.

Reviewer #2 (Comments to the Authors (Required)):

Structural characterization and inhibition of the interaction between cg-TOG and TACC2

In this manuscript, Shelford et al. investigate the interaction between the TOG domain protein ch-TOG and TACC3. These proteins are components of several common complexes, such as TACC3-clathrin-ch-TOG-GTSE1, which is involved in microtubule cross-linking, and TACC3-ch-TOG, which tracks microtubule plus ends. They also have independent functions; for example, ch-TOG binds to kinetochores via HEC1. The authors highlight the complexity of understanding the specific roles of TACC3 and ch-TOG within different complexes.

To elucidate the structure of the TACC3-ch-TOG complex, the authors conducted extensive biophysical and structural analyses of both proteins individually and in combination. Using a broad range of experimental techniques, including NMR, AUC, EPR, MST, and HDX-MS, they obtained detailed structural insights. These include the identification of the parallel coiled-coil dimer of TACC3 (residues 629-838) and the helical core of the ch-TOG region (residues 1817-1957), as well as the formation of the TACC3-ch-TOG complex.

The Affimer strategy introduced in this study specifically targets TACC3 TD and selectively disrupts the interaction between TACC3 and ch-TOG both in vitro and in vivo. The researchers identified six different Affimer groups, further characterizing E4, E5, E7, and E8. E4, E7, and E8 were found to bind to amino acids 670-682 of TACC3, which corresponds to the ch-TOG binding site. Among these, E8 exhibited the strongest binding to TACC3 and inhibited the formation of the TACC3-ch-TOG complex. When applied to cells, the Affimer disrupted the interaction between TACC3 and ch-TOG, leading to the mislocalization of ch-TOG from the mitotic spindle without affecting clathrin or TACC3. This allowed the authors to investigate the function of ch-TOG in the mitotic spindle without disrupting TACC3 localization or impairing general ch-TOG functions, which would occur when using a ch-TOG degen. Interestingly, Affimer expression also affected the localization of pericentrin, resulting in additional pericentrin foci in mitotic cells, indicating PCM fragmentation during mitosis without affecting the centrioles (as evidenced by the lack of centrin 1 in the additional pericentrin dots).

This manuscript presents complex data from a wide range of experiments in a competent and well-discussed manner. The Affimer approach is very interesting and the authors provide a very good case for the use of this strategy. The authors demonstrate that the TACC3-ch-TOG complex in the mitotic spindle is disrupted by Affimer expression. However, the transition to PCM fragmentation is abrupt and lacks a clear connection. This section needs improvement to enhance the flow and coherence. The pericentrin fragmentation phenotype is interesting; however, the underlying molecular mechanisms are not addressed. The authors should provide additional data in this direction. Additionally, I have several concerns outlined below that should be addressed.

We thank the Reviewer for their time and comments on our manuscript. We were very pleased with their positive assessment of our work and the Affimer approach in general.

Major concerns

1. It is impressive that authors applied a broad range of different techniques and spent quite some effort to describe the structure characterization of both TACC3 TD and ch-TOG 1817-1957 region in detail. This may be attractive to a special group of people in the field but could also have the potential risk to lose the interest of the board readers, especially given that these data are consistent with the previous models (based AlphaFold2 and deletion analysis) and also not considered as the major focus of this study. Therefore, it makes sense if the authors shorten this result section and combine the first three sections.

We appreciate the reviewer's comments regarding the extensive structural characterization of TACC3 and ch-TOG in the results. The AUC and EPR analysis of TACC3 is required to confirm the multimeric state and orientation of the TACC3 protomers in the dimer, neither of which have been described previously. The paper additionally presents – for the first time – the structure of ch-TOG 1817-1957 which we have determined by NMR spectroscopy. Therefore, the experimental results described in the manuscript are novel, and key to generating and validating an accurate structural model of the TACC3-ch-TOG complex, which we present later in the results section: these results are therefore essential to the paper and have been retained in the revised submission.

The reviewer refers to previous models, “based AlphaFold2 and deletion analysis” suggesting there is a duplication in results. However, we have not published AlphaFold2 analysis before and the publicly available AlphaFold model of TACC3 (<https://www.alphafold.ebi.ac.uk/entry/Q9Y6A5>) presents the protein as a monomer, not a parallel, coiled-coil dimer as we experimentally derive in the paper. We have previously used deletion analysis to map the ch-TOG binding site on TACC3 and this work is cited in the manuscript where we present data confirming that the Affimers bind at the same site as ch-TOG.

2. The fragmentation of centrosome PCM caused by the disruption of TACC3 and ch-TOG interaction via Affimers treatment is a relatively novel phenotype and one of the key findings in this study. However, the direct connection between the TACC3 and ch-TOG interaction and the PCM integrity during mitosis is missing. Is the TACC3-ch-TOG complex also detected in the PCM or centrosome region during mitosis? Would the recruitment and/or activation of γ -TuRC in PCM also impaired upon the Affimers treatment? More related discussion or interpretation will also help to better understand this interaction.

*The question of whether the TACC3–ch-TOG complex is detected at the PCM or centrosome was also raised by Reviewer 1. The localization of TACC3 and ch-TOG during mitosis is complicated, and we have previously published work that addresses this question (Gutiérrez-Caballero et al. 2015). Leaving aside the localizations on kinetochores, spindle microtubules and microtubule plus-ends, focusing only on the spindle pole, ch-TOG is found at the centrosome while TACC3 is in the region surrounding the PCM. As part of the revisions for this paper we looked again at these localizations and came to the same conclusion as we had previously published (please see **Reviewer Figure 1** and description above). Because these results are already published, we have not included them in the paper. Instead, we now show averaged images of the TACC3/ch-TOG localization in Figure 5 to highlight that ch-TOG and TACC3 have distinct localizations in mitotic cells.*

3. Across the study, the truncated version of TACC3, TACC3 629-838 Δ 699-765 (TACC3 TD Δ), is used for the complex structure prediction, Affimer generation, and several in vitro binding assays. It is worth having a more detailed introduction of this truncated version and why it was chosen in this study.

An explanation for the use of TACC3 629-838 Δ 699-765 has been inserted into the manuscript at the first mention of this construct within ‘Interaction of ch-TOG 1817-1957 with TACC3’.

4. The Affimer is crucial in this study to target TACC3 and disrupt the interaction between TACC3 and ch-TOG. Authors did extensive characterization of these Affimers both in vitro and in vivo, but it also raised some concerns, like the Affimer could not co-localized well with endogenous TACC3 very well in vivo (but works better with the overexpressed TACC3), the selectivity of Affimers in cell (which could be further addressed by doing the cell lysate pulldown using resin-immobilized Affimer), and decrease of TACC3 spindle recruitment upon Affimers treatment (Figure 5C-D, which may also indicate some other functions of these Affimers in cells). Authors should also address (could be non-experimentally) the pros and

cons of Affimer approach compared to some other wilder used methods, like partial deletion of the binding site or introducing the binding deficient mutants for either protein.

We have included the following sentence in the Discussion where we cover the usage of Affimers as reagents to disrupt protein-protein interactions. "While similar results can be achieved by specific mutation of the binding site, the advantage of using Affimers for disruption is that they have the potential to be deployed to cells without the need for introducing such mutations." The Affimers we report couldn't be used in the way that nanobodies can, i.e. to localize the endogenous (not overexpressed) TACC3 and to cause the inducible mislocalization of TACC3. We now acknowledge this directly in the Results section by stating "Although Affimers have the potential to be used akin to nanobodies, none of the Affimers performed well as reagents for visualization or inducible relocation of endogenous TACC3 (Figure S4)." We believe that higher affinity reagents could be used in this, however the ideal reagents would not target and occlude an important binding site. So in reality a suite of Affimers that can perform different tasks would be ideal. We hope that the edits clarify this point.

5. In Figure 4A, Affimer is used to be immobilized on the resin and pulldown other proteins, but its signal in the first E7 co-precipitation sample is much lower compared to others. Meanwhile, some extra bands (I would assume they are minus impurities from protein purification) could be observed in the TACC3 TD sample. Some special labels of these bands and/or discussion in the text would be appreciated. Later in the same experiment, TACC3 TD is used to pulldown ch-TOG as a positive control, but based on the previous experimental settings (a. The Affimers were generated based on TACC3 TD Δ . b. in the early part of the same experiment, only TACC3 TD Δ was used together with ch-TOG), TACC3 TD Δ should also be included and tested its interaction with ch-TOG.

*We thank the reviewer for their observations. In response, we repeated the co-precipitation assay in **Figure 4A** with freshly purified protein stocks (to remove the presence of impurity bands), ensured equal protein usage in the reactions and added the requested TACC3 TD Δ -ch-TOG reaction.*

6. From the current data, the phenotype of Affimer treatment is similar to the depletion of ch-TOG, which indicates that the ch-TOG-TACC3 complex is important for PCM integrity. Authors could check if Affimer treatment affects the ch-TOG level in general (using immunoblot) and around in PCM specifically (using immunofluorescence).

*Thank you for raising this point. Affimer treatment is similar but not identical to ch-TOG depletion, the main phenotype of which is multipolar spindles. Nonetheless it is important to confirm the ch-TOG levels following Affimer expression. **Figure 4G** now shows that the expression of ch-TOG (and TACC3) are not affected by Affimer expression. This is important to show next the the new PLA data showing the reduced interaction of TACC3 and ch-TOG which could be affected by expression. As part of the revisions, we also measured ch-TOG levels using microscopy and found no difference in cells expressing mCherry-tagged Affimers. We have not included this in the manuscript for space reasons given the Western blot shows the same thing.*

Minor comments

1. Fig. 4D does not show inhibition of ch-TOG binding to TACC3 in vitro as stated in the text.

We have clarified the statement regarding Figure 4D in the text.

2. Since the author put the gene name of ch-TOG, CKAP5, in the keyword, it is better to have a brief introduction of it in the text for the general audience.

In the Introduction, when ch-TOG is first mentioned, it is now described as ch-TOG/CKAP5.

3. The construct information of NusA-TACC3 TACC domain is missing in the methods section.

NusA-TACC3 TACC domain is generated by expression of the TACC3 coding sequence in pETM6T1 (as stated in the methods section). This generates a protein with a N-terminal double-His-NusA tag.

4. In Figure 1A, the second label on the left (ch-TOG 1467-2032/NusA) is supposed to indicate two different bands on two different gels, but might be a little bit misleading, it might be better to separate the label for each gel respectively.

The suggested edit to Figure 1A has been done.

5. A consistent color-coding for the five helices of ch-TOG 1817-1957 in all of the figures (especially in figures 1, 2, and 3) could be helpful to deliver the information to the readers.

We have retained the original color-coding for the structure Figures. There is a consistent color-coding between Figures 2-4 for the top scoring NMR model of ch-TOG 1817-1957. The different coloring in Figure 1 is required to show the structural variation between the top 20 NMR models for ch-TOG 1817-1957. The coloring used for the AlphaFold model score in Figure 3 and HDX analysis in Figure 4 are standard for presentation of these types of data.

6. In Figure 3A, the label of the y-axis could be changed to CSPs (ppm), which might be more friendly to the general audience. The 1* and 2* labels in the spectrum also require further description in the legend.

We have retained the original y-axis label description as it indicates the use of the NMR delta scale for chemical shift measurements. The legend for Figure 3A describes how the data presented represents chemical shift perturbations in ch-TOG residues on binding of TACC3. The legend has been updated to explain the 1 and 2* labels.*

7. The axis labels are missing for Figure S5.

The axis labels have been added to Figure S5.

8. Did the author also try to predict the structure of ch-TOG 1817-1957 region alone using AlphaFold? Does H5 more prefer to dock on the H1-H4 core or rather flee out freely?

As requested by the reviewer, we used AlphaFold2 & 3 to generate the structure of ch-TOG 1817-1957. The confidence scores for the H1-H5 helices were significant and are in close agreement with the NMR structure resolved for this protein. However, the structure prediction scores for the long loop region between the H1-H4 core and H5 were of very low confidence meaning the AlphaFold models should not be used to discuss the positioning of H5. We would refer the reviewer instead to the NMR experiments described in 'Conformation and stability of ch-TOG 1817-1957' where we show H5 is in equilibrium between 'a stable α -helical state, in which it is associated with the rest of the domain, and a dissociated state in which it is partially unfolded.'

9. In Figure S7, based on the legend, the panel D is missing.

This was an error. However, this Figure (now Fig S4) has been reorganized.

10. In Figure 5B, the representative images of mCherry control don't look very consistent with the quantification results (Fig 5C), so it would be better to update with another image.

We couldn't see what the Reviewer's concern was here and have left this panel intact.

11. In Figure S8, further quantification is required to demonstrate clathrin is not impacted by the Affimers treatment.

Quantification of the spindle recruitment of TACC3 and clathrin has been added to this Figure which is now Figure S4.

Reviewer #3 (Comments to the Authors (Required)):

In this manuscript, Shelford and colleagues focus on two important regulators of spindle assembly, TACC3 and ch-TOG. First, they perform a structural characterization of the complex, providing structural and molecular detail for the TACC3/ch-TOG interaction. Building on these bases, they identify Affimers with the ability to bind TACC3 and interfere with ch-TOG interaction. They go on characterizing the cellular effects of identified Affimers, in order to dissect the mitotic functions that are directly dependent on the TACC3/ch-TOG complex. The work addresses an important point, since dissecting the roles of specific subpools of multi-functional proteins such as TACC3 and ch-TOG is an interesting and challenging objective. Authors use approaches going from computational, biochemical and biophysical to cellular assays, experimental data are solid and overall clearly presented. Still, I recommend additional experiments to be performed prior to publication, to strengthen the conclusions and widen the obtained information. Specific points are listed below.

We thank the Reviewer for their very helpful comments on our paper. It was great to hear that our overall approach in the manuscript was appreciated by the Reviewer.

1. The manuscript describes a structural model of the interface between TACC3 and ch-TOG, generated using AlphaFold2-Multimer. However, this approach has several limitations that should be addressed, in particular: a) While AlphaFold2 was a groundbreaking tool in protein structure prediction, AlphaFold3 has since been released and has demonstrated superior performance in predicting protein structures, especially in multimeric contexts. The use of AlphaFold2 instead of AlphaFold3 might result in less accurate or less reliable models. The authors should consider updating their methodology to incorporate the latest version to enhance the accuracy and confidence in their predictions. b) The predictions are based on specific fragments of TACC3 (residues 629-838 Δ 699-765) and ch-TOG (residues 1817-1957). While focusing on fragments can simplify the modeling process and reduce computational load, it may overlook important interactions and conformational changes that occur in the context of the full-length proteins. Full-length protein analysis, at least at the computational level, would provide a more comprehensive view of the interface and its functional implications.

We have performed AlphaFold2 & 3 modelling of the TACC3-ch-TOG complex and found that the model generated using AlphaFold2-Multimer displayed higher confidence scores than the one produced using AlphaFold3. Therefore, we have retained the AlphaFold2 model in manuscript.

As requested by the reviewer, we have carried out AlphaFold3 modeling for the full-length TACC3-ch-TOG complex. The confidence scores for the TACC3 TACC domain (residues 629-838) are high, but the prediction for the N-terminal region is very low. For full-length ch-TOG, the confidence scores are variable throughout the protein. The server generates a top model for the complex with an ipTM score of 0.29, which indicates a failed prediction. We have therefore, not presented this data in the manuscript. It is worth noting, however, that even in this broadly incorrect model, the interaction mode between ch-TOG and TACC3 through H5 and the mapped region of TACC3 is retained.

2. The Affimer binding to TACC3 and the impaired interaction between TACC3 and ch-TOG are key results on which the interpretation of cellular assays is based. As such, I find the experiment in Fig. 4E is not sufficiently informative. Furthermore, I find confusing that Affimers E7 and E8 display (Fig. 4E) such different binding ability to TACC3 and different capability to interfere with ch-TOG binding to TACC3, but highly similar phenotypes in the following assays. a. The disruption of the TACC3/chTOG complex in single cells at specific mitotic stages should be demonstrated by isPLA, or other proximity-based, assays and endogenous components. This would provide a measure of how many mitotic cells, and to which extent, display reduced interaction. Parallel PLA assays using mCherry and TACC3 (or GFP) antibodies would enable authors to evaluate and compare to which extent and

where the Affimers/target interaction occurs, since localization patterns are not too clear (S7). b. Dose response assays (with different amounts of transfected Affimers would also be useful for a better understanding of Affimers mode of action.

In response to the Reviewer's comments, we performed in situ proximity ligation assays to quantify the effect of mCherry-Affimer expression on TACC3-ch-TOG complex formation. This data is presented in Figure 4E & F and replaces the original co-precipitation assay. The isPLAs show expression of Affimers E4, E7 and E8 result in a significant reduction in TACC3-ch-TOG foci compared to a mCherry control vector indicating that the Affimers in this study are effective inhibitors of TACC3-ch-TOG complexation, and are therefore suitable tools with which to dissect the function of the TACC3-ch-TOG interaction in cell-based assays.

3. Although the TACC domain is expected not to be directly involved in the mechanisms related to the Aurka/TACC3 interplay (Aurka activation, TACC3 phosphorylation), cellular scenarios may be more complex, and I feel it would be important to show that Aurka activation (and potentially TACC3 phosphorylation) is not modified in cells upon Affimers transfection. Related to this, it is not clear to me how authors explain the decrease of TACC3 at the spindle.

In response to Reviewer 1 (point 6), the data in Fig 5C is presented as a SuperPlot and the resulting statistics show that TACC3 spindle enrichment for E4, E7 and E8, compared to mCherry, have p values of 0.333, 0.052 and 0.123 respectively. So although the enrichment appears lower, this observation is not significant, and could be due to chance. The spindle recruitment of TACC3 is significantly higher than the MLN8237 control, so ultimately, TACC3 remains on the spindle. We have changed the text on this point.

4. It appears that clathrin localization is an important result to show the specificity of action of the Affimers towards the TACC3/ch-TOG interaction and function. Images in Fig. S8 should therefore be accompanied by a quantification analysis.

Quantification of the spindle recruitment of TACC3 and clathrin has been added to this Figure which is now Figure S4.

5. Some additional quantifications under the experimental conditions of Figures 5-6 would help interpreting data and add relevant information: a. how is the cell-to-cell variability of the mCherry Affimer signal? b. which is the amount of spindle tubulin upon treatment? c. Is ch-TOG localization altered in prometaphase too?

To address this point, we have measured the intensity of mCherry or mCherry-tagged Affimers in mitotic cells and tested whether that correlated with phenotype (2 versus >2 PCNT foci). While there is cell-to-cell variability, i) there is no consistent differences in expression of the different affimers, ii) there was no correlation with severity of phenotype across three experimental repeats (Reviewer Figure 2).

Reviewer Figure 2

To answer point b, we have added a new panel (Figure 5D) where the amount of spindle tubulin is quantified. It shows no difference between mCherry or mCherry-tagged Affimers.

For point c, we have not addressed this explicitly with new data but can confirm that the chTOG localization is altered in prometaphase cells. Our focus here was on metaphase in order to standardize the analysis.

6. Authors propose force unbalances to underlie to observed fragmentation. Rescue experiments by altering spindle forces in independent manners would support this hypothesis and would add information about the mechanism through which this chTOG/TACC3 function is exerted.

*We have tested this point by examining whether the PCNT foci frequency in cells expressing Affimers is reduced by nocodazole. We found no evidence for rescue. We used tubulin staining to confirm that this treatment was sufficient to depolymerize microtubules. Additionally, we could see a change in the centriolar PCM arrangement in nocodazole-treated cells so the treatment was apparently sufficient to decrease forces on the poles. These results are now in **Figure S5B** and are described in the text. We have removed the sentence in the Discussion referring to the fragmentation being caused by microtubule forces.*

7. Are Affimers able to bind TACC1 and TACC2, and can this contribute to the phenotypes?

We were unable to test whether the Affimers bind TACC1 and TACC2 in vitro because recombinantly expressed TACC1 protein is proteolytically unstable, and TACC2 proteins do not express solubly. However, while there are large regions within the TACC domains of these proteins that are highly conserved, and which define this protein family, the site on TACC3 to which we have mapped binding of chTOG and the Affimers falls outside these conserved regions, being only 33% conserved with the other family members, therefore cross-reactivity is unlikely.

Minor

1. Fig. 6E. If I correctly understand the display mode of the plot, there are 5 cells with 2 centrosoles in each spot, out of the 21 with supernumerary PCM spots (Affimer 8). Would this imply that both PCM fragmentation and centrosome amplification occur, although to different extents?

This was a very insightful comment. The Reviewer's reading of the plot is correct. However, due to the low numbers of cells it is hard to say if this fraction is meaningful. In E4 for example there happen to be 2 such cells from 9 with supernumerary PCM spots; which is a similar rate to that seen in E8.

2. Could authors explain in more detail how the time lapse analysis is performed, in particular how "transitions" are defined?

The following definition has been added to the Methods section: " Transitions were defined: G2–Prometaphase, nuclear envelope breakdown; Prometaphase–Metaphase, all chromosomes approximately aligned at the metaphase plate; Metaphase–Anaphase, chromosome segregation."

3. A recent paper (Rajeev et al., 2023, mentioned in the Introduction) has addressed this question by a different approach, similar to the one used in Fig 7. Authors should comparatively comment the results observed by the different approaches.

Although at first glance the two studies are similar, there are some subtle differences between Rajeev et al. 2023 and our paper. Rajeev et al. use a TACC3 knockdown-replacement system developed by our lab to uncover a potential role for TACC3 in g-TURC recruitment to the centrosome. They describe TACC3 as a centrosomal adaptor, but we don't see TACC3 at centrosomes (described above). So it is a little difficult to make a statement comparing our studies.

February 3, 2025

RE: JCB Manuscript #202407002R

Stephen Royle
University of Warwick

Dear Prof. Royle:

Thank you for submitting your revised manuscript entitled "Structural characterization and inhibition of the interaction between ch-TOG and TACC3". We appreciate your patience as we arrived at a decision, which was delayed due to editor availability. As you will see, all reviewers are satisfied and recommend publication. Please consider their comments to further improve the work via changes to the text. Reviewer 3 requests that PCM fragmentation is quantified in Fig 5S as in Fig 6 (point 3). We appreciate the reviewer's interest in consistency but we leave this to your discretion. We would be happy to publish your paper in JCB pending final revisions necessary to meet our formatting guidelines (see details below).

A. MANUSCRIPT ORGANIZATION AND FORMATTING:

Full guidelines are available on our Instructions for Authors page, <http://jcb.rupress.org/submission-guidelines#revised>. Submission of a paper that does not conform to JCB guidelines will delay the acceptance of your manuscript.

1) Text limits: Character count for Articles is < 40,000, not including spaces. Count includes abstract, introduction, results, discussion, and acknowledgments. Count does not include title page, figure legends, materials and methods, references, tables, or supplemental legends.

2) Figures limits: Articles may have up to 10 main figures and 5 supplemental figures/tables.

3) Figure formatting: Scale bars must be present on all microscopy images, including inset magnifications. Molecular weight or nucleic acid size markers must be included on all gel electrophoresis. Please avoid pairing red and green for images and graphs to ensure legibility for color-blind readers. If red and green are paired for images, please ensure that the particular red and green hues used in micrographs are distinctive with any of the colorblind types. If not, please modify colors accordingly or provide separate images of the individual channels.

** Please ensure a scale bar on Figure S4E is visible.

4) Statistical analysis: Error bars on graphic representations of numerical data must be clearly described in the figure legend. The number of independent data points (n) represented in a graph must be indicated in the legend. Statistical methods should be explained in full in the materials and methods. For figures presenting pooled data the statistical measure should be defined in the figure legends. Please also be sure to indicate the statistical tests used in each of your experiments (either in the figure legend itself or in a separate methods section) as well as the parameters of the test (for example, if you ran a t-test, please indicate if it was one- or two-sided, etc.). Also, if you used parametric tests, please indicate if the data distribution was tested for normality (and if so, how). If not, you must state something to the effect that "Data distribution was assumed to be normal but this was not formally tested."

5) Abstract and title: The abstract should be no longer than 160 words and should communicate the significance of the paper for a general audience. The title should be less than 100 characters including spaces. Make the title concise but accessible to a general readership.

6) Materials and methods: Should be comprehensive and not simply reference a previous publication for details on how an experiment was performed. Please provide full descriptions in the text for readers who may not have access to referenced manuscripts. We also provide a report from SciScore and an associate score, which we encourage you to use as a means of evaluating and improving the methods section.

** Please include brief descriptions of protein purifications and co-precipitation, MTSL protein labeling, NMR spectroscopy protein expression and analysis. Please provide a full description of affimer screening against TACC3 peptides.

7) Please be sure to provide the sequences for all of your primers/oligos, plasmids, and RNAi constructs in the materials and methods. You must also indicate in the methods the source, species, and catalog numbers (where appropriate) for all of your antibodies. Please also indicate the acquisition and quantification methods for immunoblotting/western blots.

8) Microscope image acquisition: The following information must be provided about the acquisition and processing of images:
a. Make and model of microscope

- b. Type, magnification, and numerical aperture of the objective lenses
- c. Temperature
- d. Imaging medium
- e. Fluorochromes
- f. Camera make and model
- g. Acquisition software
- h. Any software used for image processing subsequent to data acquisition. Please include details and types of operations involved (e.g., type of deconvolution, 3D reconstitutions, surface or volume rendering, gamma adjustments, etc.).

10) Supplemental materials: There are strict limits on the allowable amount of supplemental data. Articles may have up to 5 supplemental figures. Please also note that tables, like figures, should be provided as individual, editable files. A summary of all supplemental material should appear at the end of the Materials and methods section.

13) ORCID IDs: ORCID IDs are unique identifiers allowing researchers to create a record of their various scholarly contributions in a single place. At resubmission of your final files, please provide an ORCID ID for all authors.

15) A data availability statement is required for all research article submissions. The statement should address all data underlying the research presented in the manuscript. Please visit the JCB instructions for authors for guidelines and examples of statements at (<https://rupress.org/jcb/pages/editorial-policies#data-availability-statement>).

Please note that JCB requires authors to submit Source Data used to generate figures containing gels and Western blots with all revised manuscripts. This Source Data consists of fully uncropped and unprocessed images for each gel/blot displayed in the main and supplemental figures. Since your paper includes cropped gel and/or blot images, please be sure to provide one Source Data file for each figure that contains gels and/or blots along with your revised manuscript files. File names for Source Data figures should be alphanumeric without any spaces or special characters (i.e., SourceDataF#, where F# refers to the associated main figure number or SourceDataFS# for those associated with Supplementary figures). The lanes of the gels/blots should be labeled as they are in the associated figure, the place where cropping was applied should be marked (with a box), and molecular weight/size standards should be labeled wherever possible. Source Data files will be directly linked to specific figures in the published article.

WHEN APPROPRIATE: The source code for all custom computational methods published in JCB must be made freely available as supplemental material hosted at www.jcb.org. Please contact the JCB Editorial Office to find out how to submit your custom macros, code for custom algorithms, etc. Generally, these are provided as raw code in a .txt file or as other file types in a .zip file. Please also include a one-sentence summary of each file in the Online Supplemental Material paragraph of your manuscript.

Journal of Cell Biology now requires a data availability statement for all research article submissions. These statements will be published in the article directly above the Acknowledgments. The statement should address all data underlying the research presented in the manuscript. Please visit the JCB instructions for authors for guidelines and examples of statements at (<https://rupress.org/jcb/pages/editorial-policies#data-availability-statement>).

B. FINAL FILES:

Thank you for your attention to these final processing requirements. Please revise and format the manuscript and upload materials within 7 days. If you need an extension for whatever reason, please let us know and we can work with you to determine a suitable revision period.

Thank you for this interesting contribution, we look forward to publishing your paper in Journal of Cell Biology.

Sincerely,

Monica Bettencourt-Dias
Monitoring Editor
Journal of Cell Biology

Tim Fessenden
Scientific Editor
Journal of Cell Biology

Reviewer #1 (Comments to the Authors (Required)):

The authors have addressed my main concerns. Please address also the following remaining minor points:

-How is it that, in Figure 5B, in the control condition where cells express mCherry alone, there is mCherry signal in the spindle?

Authors:

The spindle can be infiltrated by mCherry which is only 27 kDa. We often see a slight enrichment of GFP, mCherry or other fluorescent proteins in the spindle region.

New comment:

In that case, the authors should mention this in the manuscript, since it is confusing to see spindle signal for both mCherry control and affimers.

-All IF figures: to be consistent, for each channel I would indicate the antibody used on one side and the expressed construct on the other side of the figure. Currently this is inconsistent and confusing in several cases, e.g. Fig. 7A (GFP labeled as chTOG-GFP). Fig. S7A,B,C (various inconsistencies)

Authors:

Again, thanks for spotting this. Corrected.

New comment:

The example "e.g. Fig. 7A (GFP labeled as chTOG-GFP)" has not been corrected.

Also, in Fig. S7B, the caption only describes the quantification of PCNT foci, while it should also describe the representative images showed at the left. On top of these images, including the antibody used (PCNT) would also help with comprehension of the panel.

-About the paragraph in the Results section that talks about Supplementary Figure S7: "None of the Affimers performed well as reagents for visualization or inducible relocalization of TACC3 (Supplementary Figure S7). However, we were able to see colocalization of each of the three Affimers with overexpressed GFP-TACC3 (Supplementary Figure S7)." I believe these experiments need to be explained in a bit more detail (such as the role of rapamycin) and the significance of the outcome.

Authors:

The Reviewer is correct that the description of the results was too terse. Essentially Affimers could be used like nanobodies such that we could use them to i) visualise the endogenous TACC3 or ii) mislocalize TACC3 by inducibly relocalizing the Affimer. Neither of these things was possible, although when GFP-TACC3 was over-expressed, we could use the Affimers to label it. We have changed the sentence to read "Although Affimers have the potential to be used akin to nanobodies, none of the Affimers performed well as reagents for visualization or inducible relocalization of endogenous TACC3 (Figure S4A-C)." The section has been reworked because of other changes and now the colocalization result precedes this sentence, which should make it all clearer.

New comment:

Our main point was to provide more detail on the "relocalization" experiment. What is this experiment, how was it done and why? What does rapamycin do here etc?

Reviewer #2 (Comments to the Authors (Required)):

The authors have addressed my comments, as well as those of the other reviewers, significantly enhancing the manuscript's quality to a publishable standard. However, the discussion remains relatively brief and lacks sufficient conclusiveness regarding PCM fragmentation and the role of the TACC3-ch-TOG complex in maintaining PCM integrity.

In the introduction, the authors reference the study by the Lüders lab (Ali et al., 2023). Including a short discussion of this work in the manuscript's discussion section would be valuable. Ali et al. demonstrated that ch-TOG depletion impacts the gamma-TuRC without affecting PCNT. Additionally, the absence of TACC3 at the mitotic PCM, as reported by Gutierrez-Caballero et al. (2015), warrants mention in the discussion. Expanding on these points would provide greater depth and context, thereby strengthening the overall manuscript.

Ultimately, the mechanism by which TACC3-ch-TOG impacts the PCM remains unresolved and will require further investigation.

Reviewer #3 (Comments to the Authors (Required)):

In the revised version of the manuscript, Shelford and colleagues have addressed some of the issues I had raised, as the Editor suggested, including new results or controls/quantifications in revised Figures and Supplementary material. They also added Figures/Information for reviewers only, that for different reasons were not added to the revised manuscript. I have a few minor points that I feel should be addressed before publication:

1. In Figure 4E, the E7, E8 and ch-TOG PLA panels appear different from the other conditions not only in relation to the specific signals but also in terms of background. I would recommend authors to make an additional check on the images.
2. In the time lapse Transitions definitions added to the Methods, I would stick to information that can be directly derived from the used markers (g-tub and DNA), therefore I would not refer to nuclear envelope breakdown (line 771 "G2-Prometaphase, nuclear envelope breakdown"); in addition, there is a typo in the added text (line 773 "metphase")
3. Authors have added an experimental setting using nocodazole (FigS5B) to investigate whether eliminating microtubules would rescue the PCM fragmentation phenotype by removing spindle-associate forces, concluding that this is not the case. I find confusing that the results of the nocodazole experiments are shown with a different graphical layout with respect to the phenotype to be potentially rescued (Fig. 6B). The classification in "2" or ">2" PCNT foci shown in Fig. 6B should be used also in S5B and the % fraction of cells with >2 PCNT foci should be added for all conditions, with the statistical significance. If there is a need for introducing the additional classification of "normal", "expanded" and "fragmented" only to the nocodazole experiment, this is an additional information to be shown separately and commented on. Also, it would be important to include the protocol details (I find Nocodazole concentration, but I cannot find the time of treatment, also with respect to the transfection).
4. I would like to add a comment on a sentence added in response to Reviewer 1. The added result showing that the PCM fragmentation phenotype is not observed in HEK293 cells is somehow limiting the significance of the identified role of the

chTOG/TACC3 interaction. Authors have inserted a sentence to comment on this finding; concerning it, I am not sure about the definition of HeLa as "cells with altered PCM". Maybe a more general statement about "cancer or non-cancer", which was the original input from Reviewer 1, would be more adherent to the tested conditions (although having only 1 cancer and 1 non-cancer with contrasting results does not allow for generalisation).